



# The Great Lakes Runoff Intercomparison Project Phase 4: The Great Lakes (GRIP-GL)

Juliane Mai[*,1], Hongren Shen[*,1], Bryan A. Tolson[*,1], Étienne Gaborit[*,2], Richard Arsenault[3], James R. Craig[1], Vincent Fortin[2], Lauren M. Fry[4], Martin Gauch[5], Daniel Klotz[5], Frederik Kratzert[5,6], Nicole O'Brien[7], Daniel G. Princz[8], Sinan Rasiya Koya[9], Tirthankar Roy[9], Frank Seglenieks[7], Narayan K. Shrestha[7], André G. T. Temgoua[7], Vincent Vionnet[2], and Jonathan W. Waddell[10]

[*]Lead contributors. Other authors are ordered alphabetically by last name.
[1]Department of Civil and Environmental Engineering, University of Waterloo, Waterloo, ON, Canada.
[2] Meteorological Research Division, Environment and Climate Change Canada, Dorval, QC, Canada.
[3]Department of Construction Engineering, École de technologie supérieure, Montreal, QC, Canada.
[4]Great Lakes Environmental Research Laboratory, National Oceanic and Atmospheric Administration, Ann Arbor, MI, USA.
[5]Institute for Machine Learning, Johannes Kepler University, Linz, Austria.
[6]Google Research, Vienna, Austria.
[7]National Hydrological Service, Environment and Climate Change Canada, Burlington, ON, Canada.
[8]National Hydrological Service, Environment and Climate Change Canada, Saskatoon, SK, Canada.
[9]Department of Civil and Environmental Engineering, University of Nebraska–Lincoln, Lincoln, NE, USA.
[10]Great Lakes Hydraulics and Hydrology Office, U.S. Army Corps of Engineers, Detroit, MI, USA.

**Correspondence:** Juliane Mai (juliane.mai@uwaterloo.ca)





**Abstract.** Model intercomparison studies are carried out to test and compare the simulated outputs of various model setups over the same study domain. The Great Lakes region is such a domain of high public interest as it not only resembles a challenging region to model with its trans-boundary location, strong lake effects, and regions of strong human impact but is also one of the most densely populated areas in the United States and Canada. This study brought together a wide range of researchers

setting up their models of choice in a highly standardized experimental setup using the same geophysical datasets, forcings, common routing product, and locations of performance evaluation across the 1 million square kilometer study domain. The study comprises 13 models covering a wide range of model types from Machine Learning based, basin-wise, subbasin-based, and gridded models that are either locally or globally calibrated or calibrated for one of each of six predefined regions of the watershed. Unlike most hydrologically focused model intercomparisons, this study not only compares models regarding their

capability to simulated streamflow (Q) but also evaluates the quality of simulated actual evapotranspiration (AET), surface soil moisture (SSM), and snow water equivalent (SWE). The latter three outputs are compared against gridded reference datasets. The comparisons are performed in two ways: either by aggregating model outputs and the reference to basin-level or by regridding all model outputs to the reference grid and comparing the model simulations at each grid-cell.

The main results of this study are: (1) The comparison of models regarding streamflow reveals the superior quality of the

Machine Learning based model in all experiments performance; even for the most challenging spatio-temporal validation the ML model outperforms any other physically based model. (2) While the locally calibrated models lead to good performance in calibration and temporal validation (even outperforming several regionally calibrated models), they lose performance when they are transferred to locations the model has not been calibrated on. This is likely to be improved with more advanced strategies to transfer these models in space. (3) The regionally calibrated models – while losing less performance in spatial and

spatio-temporal validation than locally calibrated models – exhibit low performances in highly regulated and urban areas as well as agricultural regions in the US. (4) Comparisons of additional model outputs (AET, SSM, SWE) against gridded reference datasets show that aggregating model outputs and the reference dataset to basin scale can lead to different conclusions than a comparison at the native grid scale. This is especially true for variables with large spatial variability such as SWE. (5) A multi-objective-based analysis of the model performances across all variables (Q, AET, SSM, SWE) reveals overall excellent

performing locally calibrated models (i.e., HYMOD2-lumped) as well as regionally calibrated models (i.e., MESH-SVS-Raven and GEM-Hydro-Watroute) due to varying reasons. The Machine Learning based model was not included here as is not setup to simulate AET, SSM, and SWE. (6) All basin-aggregated model outputs and observations for the model variables evaluated in this study are available on an interactive website that enables users to visualize results and download data and model outputs.



# 1 Introduction

Model intercomparison projects are usually massive undertakings especially when multiple (independent) groups come together comparing a wide range of models over large regions and a large number of locations (Duan et al., 2006; Smith et al., 2012; Best et al., 2015; Kratzert et al., 2019d; Menard et al., 2020; Mai et al., 2021; Tijerina et al., 2021). The aim of such projects is diverse. It might be to identify models that are most appropriate for certain objectives (e.g., simulating high flows or representing soil moisture) or to study differences in model setups and implementations in detail.

Intercomparison projects are also well suited to introduce new models through the inherent benchmarking with other models (Best et al., 2015; Kratzert et al., 2019d; Rakovec et al., 2019). Especially, the recent successful application of data-driven models in hydrologic applications necessitate standardized experiments– such as model intercomparison studies- making sure that the models are using the same information and strategies to enable fair comparisons.

Several model intercomparison have been carried out over subdomains (here called "regions") of the Great Lakes in the
past (Fry et al., 2014; Gaborit et al., 2017a; Mai et al., 2021) under the umbrella of the Great Lakes Runoff Intercomparison Projects (GRIP). The GRIP projects were envisioned since the Great Lakes region with its transboundary location leads to (a) challenges such as finding datasets that are consistent across borders or models that are only setup on either side of the border, (b) challenges in the modeling itself due to, for example, climatic conditions driven by large lake effects and substantial areas of heavy agricultural land use and urban areas, and (c) a high public interest since the Great Lakes watershed is one of the
most densely populated areas in Canada (32% of population) and the United States (8% of population) (Michigan Sea Grant, 2022). Another reason for high public interest is that changes in runoff to the Great Lakes have implications for water levels, which have undergone dramatic changes (from record low in 2012-2013 to record highs in 2017-2020) in recent decades. The aforementioned GRIP projects were setup over regions of the entire Great Lakes watershed, i.e. Lake Michigan (GRIP-M; Fry et al., 2014), Lake Ontario (GRIP-O; Gaborit et al., 2017a), and Lake Erie (GRIP-E; Mai et al., 2021), and primarily
investigated the model performances regarding simulated streamflow even though studies show that additional variables beyond streamflow might be helpful to improve realism of models (Demirel et al., 2018; Tong et al., 2021).

Previous GRIP studies have pointed out limitations regarding the conclusions that can be drawn. Especially, the last study (GRIP-E by Mai et al. (2021)) listed the following challenges in the concluding remarks: (I) Short period of forcings of only five years to calibrate and evaluate the models. (II) Comparison of streamflow only but no additional variables like soil
moisture. (III) Even though all models use the same forcings all other datasets (soil texture, vegetation, basin delineation, etc.) were not standardized. Hence model differences might be due to different quality input data but not by differences in models themselves. The study presented here will address these previous GRIP study limitations. Furthermore, this study demonstrates some unique intercomparison aspects that are not necessarily common practice in large-scale hydrologic model intercomparison studies. Key noteworthy aspects of this intercomparison study are as follows:

**Datasets used:** Hydrologic and land-surface models need data to be setup (e.g., soil, landcover) and to be forced with (e.g., precipitation). The forcing data have, at best, a high temporal and spatial resolution, a complete coverage of the domain of interest, a long available time period, and no missing data. In this study models will be using common datasets for





these forcings and all geophysical datasets required to setup the various kinds of models. A recently developed 18-year long forcing dataset will be employed (Gasset et al., 2021). This addresses limitation (I) and (III) above.

**Variables compared:** Models are traditionally only evaluated for one primary variable (e.g., streamflow). In this study, models will be evaluated regarding variables beyond streamflow such as evapotranspiration, soil moisture, and snow water equivalent. This is addressing the limitation (II) above.

**Method of comparison:** A model intercomparison might include various model types, e.g., lumped models vs. semi-distributed or conceptual vs. physically-based vs. data-driven. To compare these models – especially when not only comparing time
series like streamflow – model-agnostic methods are needed (meaning methods that do not favour one type of model over others). We will perform comparisons on different scales and using metrics to account for these issues. This addresses an issue associated with limitation (II).

**Types of models considered:** Model intercomparison studies are traditionally using conceptual or physically based hydrologic models. The rising popularity and success of data-driven models in hydrologic science (e.g., Herath et al., 2021;
Nearing et al., 2021; Feng et al., 2021), however, warrant the inclusion of such models in such studies given that the data-driven, machine-learning based models face the same information and input data as traditional models to guarantee a fair comparison. A machine-learning based model was included in this study demonstrating advantages and limitations of state-of-the-art Machine Learning-based models compared to conceptual and physically based models.

**Communication of results:** Large studies with many models and locations where they are compared, lead to vast amounts
of model results. During such a study, it is challenging for collaborators to process and compare all results. After the study is completed, it is equally challenging to present all results in detail in a manuscript such that the data and results are accessible to others. In this study we shared data through an interactive website, first internally accessible during the study, and then publicly, to enhance data sharing, exploring, and communication with a broader audience.

The remainder of the paper is structured as follows. Section 2 presents the materials used and methods applied including a
description of datasets, models, metrics, and types of comparative analyses. Section 3 describes and discusses the results of the analyses performed while section 4 summarizes the conclusions that can be drawn from the experiments performed here. Note that this work is accompanied by an extensive supplementary materials document, primarily providing more details for model setups, and an interactive website (Mai, 2022) for sharing and exploring comparative results.

## 2 Materials & Methods

This section first describes the study domain (Sect. 2.1), datasets used to setup and force models (Sect. 2.2) as well as the common routing product produced for this study and used by most of the models (Sect. 2.3). A brief description of the thirteen participating models can be found in Sect. 2.4. The datasets used to calibrate and validate the models regarding streamflow as well as the datasets to evaluate the models beyond streamflow are introduced in Sect.s 2.5 and 2.6, respectively, followed





by Sect. 2.7 defining the metrics to determine the models' ability to simulate the variables provided in these datasets using
two different approaches depending on which spatial aggregation is applied (Sect. 2.8). A multi-objective analysis described in
Sect. 2.9 is performed to augment the previous single objective analyses that evaluated performance across the four variables
independently. The data used (e.g., model inputs) and produced (e.g., model outputs) in this study are made available through
an interactive website, for which features are explained in Sect. 2.10.

## 2.1   Study domain

The domain studied in this work is the St. Lawrence River watershed including the Great Lakes basin and the Ottawa River
basin. The five Great Lakes located within the study domain contain 21% of the world's surface fresh water and approximately
34 million people in the United States and Canada live in the Great Lakes basin. This is about 8% of the U.S. population
and about 32% of Canada's population (Michigan Sea Grant, 2022). The study domain chosen consists of six major regions,
i.e. the five local lake regions (i.e., regions are partial watersheds that do not include upstream areas draining into upstream
lakes) draining into Lake Superior, Lake Huron, Lake Michigan, Lake Erie and Lake Ontario, and one region that is defined
by the Ottawa River watershed. The latter was included as it was important to some collaborators to evaluate how well models
performed on this highly managed/human impacted watershed. Furthermore, the Ottawa River flow has implications for Lake
Ontario outflow management.

The study domain (including the Ottawa River watershed) is 915,798 km$^2$ in size of which 630,844 km$^2$ (68.9%) are land
while 284,954 km$^2$ (31.1%) are water bodies. These estimates are derived using the common routing product used in this study;
more information about this product can be found in Sect. 2.3. The outline of the study domain is displayed in Fig. 1.

The land cover of land areas (water body excluded) is dominated by forest (71%) and cropland (6.2%) estimated based on
the North American Land Change Monitoring System (NALCMS;  NACLMS, 2017). The soil types of land areas (water body
excluded) are dominated by sandy loam (47.3%), loam (19.1%), loamy sand (11.5%), and silt loam (10.5%). These estimates
are derived using the vertically-averaged soil data based on the entire soil column up to 2.3 m from the Global Soil Dataset
for Earth System Models (GSDE; Shangguan et al., 2014) dataset and soil classes are based on the USDA classification. The
elevation of the Great lakes watershed (incl. Ottawa River watershed) ranges from 17 m to 1095 m. The mean elevation is
270 m and the median is 247 m. These estimates are derived using the HydroSHEDS 90 m (3 arcsec) digital elevation model
(DEM) data (Lehner et al., 2008). These datasets – NALCMS, GSDE, and HydroSHEDS – summarize the common datasets
used by all partners in this study to setup models. More details about the datasets are provided in Sect. 2.2.

## 2.2   Meteorologic forcings and geophysical datasets

All models contributing to this study used the same set of meteorologic forcings and geophysical datasets to setup and run
their models. Additional datasets required to setup specific models had to be reported to the team and made available to all
collaborators in order to make sure everyone would have the chance to use that additional information as well. The common
datasets used here were determined after the preceding Great Lakes Runoff Intercomparison project for Lake Erie (GRIP-E;
Mai et al., 2021) in which contributors were allowed to use datasets of their preference. The team for the Great Lakes wide



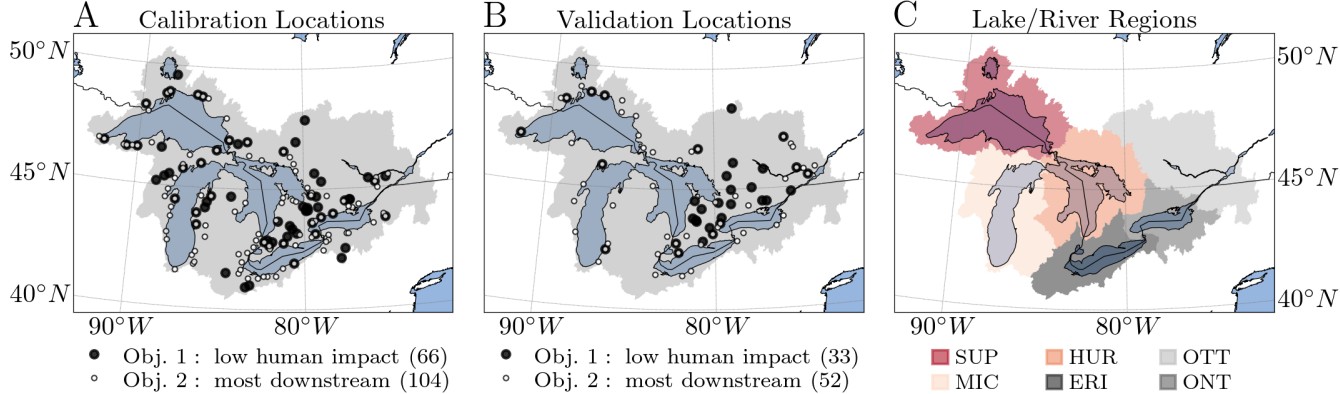

**Figure 1. Study domain and streamflow gauging locations.** In total 212 streamflow gauging stations (dots) located in the Great Lakes watershed including the Ottawa River watershed (gray area in panels A and B) have been used in this study. Note that all selected gauging stations eventually drain to one of the Great Lakes or the Ottawa River and none are downstream of any of the Great Lakes. Hence, most models do not simulate the areas of the Great Lakes themselves. Panel A shows the location of stations used for calibration regarding streamflow: 66 of them are downstream of a low-human impact watershed (objective 1; large black dots) and 104 stations are most downstream draining into one of the five lakes or the Ottawa River (objective 2; smaller dots with white center). In total, there are 141 stations used for calibration as 29 stations are both low-human impact and most downstream (large black dots with white center; $141 = 66 + 104 - 29$). Panel B shows the 71 validation stations of which 33 are low human impact, 52 are most downstream and 14 are both low human impact as well as most downstream ($71 = 33 + 52 - 14$). The number of stations is added in parenthesis to the labels in the legend. Panel C shows the six main regions of the study domain, i.e., the Lake Superior region (SUP), the Lake Michigan region (MIC), the Lake Huron region (HUR), the lake Erie region (ERI), the Ottawa River region (OTT), and the Lake Ontario watershed (ONT).

project (GRIP-GL) evaluated all datasets used by all models in GRIP-E and decided together which single dataset to commonly use in GRIP-GL. This led to the situation that most models had to be setup again as for all models at least one dataset differed between what was used in GRIP-E and decided to be used in GRIP-GL. The common datasets are briefly described in the following.

The Regional Deterministic Reanalysis System v2 (RDRS-v2) was employed as meteorologic forcing (Gasset et al., 2021). RDRS-v2 is an hourly and 10 km by 10 km meteorologic forcing dataset covering North America (Fig. 2A). Forcing variables such as precipitation, temperature, humidity, wind speed, among others are available from January 2000 until December 2017 through the Canadian Surface Prediction Archive (CaSPAr) (Mai et al., 2020b). The dataset has been provided to all the participating groups in two stages: First, only data for the calibration period (January 2000 to December 2010) were shared. Second, after finalizing the calibration of all models, the data were shared for the validation period (January 2011 until December 2017). This allowed for a true blind (temporal) validation of the models. This was possible since the RDRS-v2 dataset was not publically available when the GRIP-GL project started and the participating groups (except the project leads) did not have access to the data. The forcings were aggregated according to the needs of each model. For example, lumped models used





aggregated basin-wise daily precipitation and minimum and maximum daily temperature instead of the native gridded, hourly
inputs.

The HydroSHEDS dataset (Lehner et al., 2008) was used as the common Digital Elevation Model (DEM) for the project.
It has a 3 arcsec resolution which corresponds to about 90 m at the equator. Since the upscaled HydroSHEDS DEMs with
15 arcsec (500 m) and 30 arcsec (1 km) are consistent with the best resolution dataset, the collaborators were allowed to pick

the resolution most appropriate for their setup. The DEM was then post-processed to the specific needs of each model. For
example, some lumped models required the average elevation of each watershed as an input. The data were downloaded from
the HydroSHEDS website (https://www.hydrosheds.org), cropped to the study domain and provided to the collaborators.

The Global Soil Dataset for Earth System Models (GSDE; Shangguan et al., 2014) was used as the common soil dataset
for all models in this study. This dataset with a 30 arcsec spatial resolution (approximately 1 km at the equator) contains eight

layers of soil up to a depth of 2.3 m. The data were downloaded directly from the website mentioned in the Shangguan et al.
(2014) publication and preprocessed for the needs of the collaborators and models. For example, the data were regridded to the
grid of the meteorologic RDRS-v2 forcings as this was the grid most distributed models were setup for. Some other models
required aggregation of soil properties to specific (fixed) layers different to those distributed. The project leads also converted
the soil textures provided to soil classes as this was required by a few models. All these data products were converted to a

standardized NetCDF format and shared with the collaborators.

As a common landcover dataset for all models, the North American Land Change Monitoring System (NALCMS) product
including 19 land cover classes for North America was used. The dataset has a 30 m by 30 m resolution and is based on Landsat
imagery from 2010 for Mexico and Canada and from 2011 for the United States. The data can be downloaded from a website
(NACLMS, 2017). The data were downloaded, cropped to the study domain, and saved in a common TIFF file format. Further,

model specific postprocessing of the data included regridding to model grids and aggregation of the landcover classes to the
eight classes common in MODIS. Those datasets were saved in standard NetCDF file formats and provided to the collaborators.

## 2.3    Routing product

The GRIP-GL routing product is derived from the HydroSHEDS DEM (Lehner et al., 2008), drainage directions and flow
accumulation. All of them are in 3 arcsec (90 m) resolution. The GRIP-GL routing product is generated by BasinMaker (Version

1.0; Han, 2021), which is a set of Python-based GIS tools for supporting vector-based watershed delineation, discretization and
simplification and incorporating lakes and reservoirs into the network. This toolbox has been successfully applied for producing
the Pan-Canadian (Han et al., 2020) and North American routing product (Han et al., 2021b). More detailed information, user
manual, and tutorial examples can be found on the BasinMaker website (Han et al., 2021a).

In this study, we used the HydroSHEDS DEM, flow direction, and flow accumulation at the 3 arcsec resolution for Basin-

Maker to delineate the watersheds and discretize them into subbasins. The HydroLAKES database, which provides a digital
map of lakes globally (Messager et al., 2016), is used in the watershed delineation as well. The Terra and Aqua combined
Moderate Resolution Imaging Spectroradiometer (MODIS) Land Cover Type (MCD12Q1) Version 6 data (USGS, 2019) is



**Figure 2. Meteorologic forcing data and auxiliary datasets.** (A) The meteorologic forcings are provided by the Regional Deterministic Reanalysis System v2 (RDRS-v2; 10km; hourly) (Gasset et al., 2021). The figure shows the mean annual precipitation derived for the 18 years the dataset is available and used in this study (2000-2017). The forcing inputs are pre-processed as needed by the models, e.g., aggregated to basin averages or aggregated to daily values. (B) Actual evapotranspiration estimates and (C) surface soil moisture from the Global Land Evaporation Amsterdam Model (GLEAM v3.5b; 25 km; daily) (Martens et al., 2017) as well as (D) snow water equivalent estimates from the ERA5-Land dataset (Muñoz Sabater, 2019) (10 km; daily) have been used to evaluate the calibrated model setups. All datasets have been cropped to the study domain (black line). The actual evapotranspiration and surface soil moisture (panels B and D) are not available over the lake area (missing grid cells). The panels B-D show mean annual estimates for the auxiliary variables. The average soil moisture (panel C) is based on summer time steps only (June to October) while for the snow water equivalent annual mean only values larger than 1 mm of daily snow water equivalent were considered. In both cases, this is what was considered to evaluate model performance regarding these variables.



used in BasinMaker to estimate the Manning's n coefficient for the floodplains. The global river bankfull width and depth database developed by Andreadis et al. (2013) is used to calculate geometry characteristics of river cross sections.

This routing product is compatible with other hydrological models (i.e., runoff generating models at any resolution, either gridded or vector-based) when the Raven hydrological modeling framework (Craig et al., 2020) is used as the routing module. The routing product contains adequate information for hydrologic routing simulation, including the routing network topology, attributes of watershed, lakes and rivers, and initial estimates for routing parameters (e.g., Manning's coefficient). The key to routing arbitrary resolution spatio-temporal runoff fluxes through Raven is the derivation of grid-weights to remap fluxes

of runoff onto the routing network discretization. A workshop on how to couple the models participating in this study with the Raven framework in routing-only mode was held in October 2020. After that workshop all models except one decided to use the common routing product and route their model generated runoff through the Raven modeling framework. The GEM-Hydro-Watroute model (see list of models in Sect. 2.4 and Tab. 1) was the only model to use its native routing scheme. Preliminary calibrated versions of, for example, the subbasin-based SWAT were tested with and without Raven-based routing

across 13 individually calibrated basins. These tests confirmed that SWAT showed notably improved streamflow predictions when their water fluxes for routing were handled with Raven-based routing in comparison to their native lake and channel routing approaches (Shrestha et al., 2021).

    The common routing product includes the explicit representation of 573 lakes (214 located in calibration basins and 359 in validation basins). Small lakes with a lake area less than 5 km$^2$ were not included to achieve a balance between lake

representation and computational burden. The BasinMaker requires a threshold of flow accumulation to identify streams and watersheds. We used a flow accumulation value of 5000 for this parameter. Generally, the smaller this value, the more sub-watersheds and tributaries will be identified.

    The routing product was prepared at three equivalent spatial scales. First, the routing networks for each of the 212 gauged basins (141 calibration basins and 71 validation basins) were individually delineated. The gauged basins were discretized into

4357 subbasins (2187 for calibration basins and 2170 for validation basins) with an average size of about 220 km$^2$ (221 km$^2$ for calibration basins and 215 km$^2$ for validation basins). The 573 lake subbasins are further discretized into one land and one lake HRU per lake subbasin. Second, for models that were regionally calibrated, for ease of simulating entire regions at once, the above gauge-based routing networks were aggregated into six regional routing networks. Importantly, at all 212 gauge locations, the regional routing networks has upstream routing networks that are equivalent to the individual routing networks

for each gauge. Third, the global setup for the entire Great Lakes region was prepared but not used by any of the collaborators. More details about the streamflow gauge stations is given in Sect. 2.5.

## 2.4   Participating models

The thirteen models participating in this study are listed in Tab. 1 including (a) co-authors leading the model setups, calibration, validation, and evaluation, (b) calibration strategy, (c) routing scheme used, as well as (d) temporal and (e) spatial resolution

of the models. The models are grouped according to their major calibration strategy: The first group is the Machine-Learning based model which happens to be also the only model with a global setup (Section 2.4.1), the second group is comprised of the





seven models that are locally calibrated (Section 2.4.2) and the third group is the five models that followed a regional calibration

strategy (Section 2.4.3). The models are briefly described below including a short definition of these three calibration strategies.

More details about the models can be found in the Supplementary Material; including the lists of parameters that have been

calibrated for each model.

Please note that the model names follow a pattern: the last part of the name indicates whether the model does not require

routing ('XX-lumped'), used the common Raven routing ('XX-Raven'), or used another routing scheme (i.e, 'XX-Watroute').

Table 1: **List of participating models.** The table lists the participating models as well as the lead modelers responsible for

model setups, calibration, and validation runs. The models are separated into three groups (see italic captions in table): Machine

Learning (ML) models which are all globally calibrated, hydrologic models that are calibrated at each gauge (local calibration),

and models that are trained for each region such as the Lake Erie or Lake Ontario watershed (regional calibration). Note that the

temporal and spatial resolution of the fluxes of the land surface scheme (LSS) can be different from the resolutions used in the

routing (Rout.) component. All LSS grids are set to the RDRS-v2 meteorological data forcing grid of around 10 km by 10 km.

The two numbers given in the column specifying the spatial resolution (XXX+YYY) correspond to the spatial resolution of

the models regarding calibration basins (XXX) and validation basins (YYY).

| Model Name | Lead Modeler(s) | Calibr. Strategy | Routing Scheme | Temporal Resolution | Spatial Resolution |
|---|---|---|---|---|---|
| *Machine Learning model(s) (global calibration):* | | | | | |
| LSTM-lumped | Gauch, Klotz & Kratzert | global | none | daily | basins (141+71) |
| *Hydrologic and land-surface model(s) with calibration of each gauge individually (local calibration):* | | | | | |
| LBRM-CC-lumped | Waddell & Fry | local | none | daily | basins (141+71) |
| HYMOD2-lumped | Rasiya Koya & Roy | local | none | daily | basins (141+71) |
| GR4J-lumped | Mai & Craig | local | none | daily | basins (141+71) |
| HMETS-lumped | Mai & Craig | local | none | daily | basins (141+71) |
| Blended-lumped | Mai, Craig & Tolson | local | none | daily | basins (141+71) |
| Blended-Raven | Mai, Craig & Tolson | local | Raven | daily | LSS: subbasins (2187+2170) |
| | | | | | Rout.: subbasins (2187+2170) |
| VIC-Raven | Shen & Tolson | local | Raven | LSS: 6 h | LSS: grid (10 km) |
| | | | | Rout.: daily | Rout.: subbasins (2187+2170) |
| *Hydrologic and land-surface model(s) with calibration of entire regions (regional calibration):* | | | | | |
| SWAT-Raven | Shrestha & Seglenieks | regional | Raven | daily | LSS: subbasins (3230+2268) |
| | | | | | Rout.: subbasins (2187+2170) |
| WATFLOOD-Raven | Shrestha & Seglenieks | regional | Raven | hourly | LSS: grid (10 km) |
| | | | | | Rout.: subbasins (2187+2170) |

*Continued on next page*





| Model Name | Lead Modeler(s) | Calibr. Strategy | Routing Scheme | Temporal Resolution | Spatial Resolution |
|---|---|---|---|---|---|
| MESH-CLASS-Raven | Temgoua & Princz | regional | Raven | LSS: 30 min | LSS: grid (10 km) |
| | | | | Rout.: daily | Rout.: subbasins (2187+2170) |
| MESH-SVS-Raven | Gaborit & Princz | regional | Raven | LSS: 10 min | LSS: grid (10 km) |
| | | | | Rout.: 6 h | Rout.: subbasins (2187+2170) |
| GEM-Hydro-Watroute | Gaborit | regional | Watroute | LSS: 10 min | LSS: grid (10 km) |
| | | | | Rout.: hourly | Rout.: grid (1 km) |

### 2.4.1  Machine-learning based model

**LSTM-lumped:** One model based on Machine Learning, a Long Short-Term Memory network (LSTM), was contributed to this study. The model is called "LSTM-lumped" throughout this study. LSTM networks were first introduced by Hochreiter and Schmidhuber (1997) and used for rainfall–runoff modeling by Kratzert et al. (2018). The model was set up using the NeuralHydrology Python library (Kratzert et al., 2022). An LSTM is a Deep Learning model that learns to model relevant hydrological processes purely from data. The LSTM setup used in this study is similar to that from Kratzert et al. (2019c),
and has been successfully applied for streamflow prediction in a variety of studies (e.g., Klotz et al., 2021; Gauch et al., 2021; Lees et al., 2021a). The LSTM setup follows a global calibration strategy, which means that the model was trained for all 141 calibration stations at the same time, resulting in a single trained model for the entire study domain that can be run for any (calibration or validation) basin as soon as the required input variables are available. The model takes daily meteorological forcings and a set of climatologic and geophysical basin characteristics based solely on the common dataset (Sect. 2.2) as inputs
to produce daily streamflow predictions. Streamflow is not part of the input variables. The LSTM training involved fitting around 300,000 model parameters. This number should, however, not be directly compared to the number of parameters in traditional hydrological models because the parameters of a neural network do not explicitly correspond to individual physical properties. The training of one model takes around 2.75 hours on a single GPU. The supplementary material contains the full list of inputs and a more detailed overview of the training procedure (Sect. S.2.1). Unlike process-based hydrologic models,
the internal states of an LSTM do not have a direct semantic interpretation, but are learned by the model from scratch. In this study, the LSTM was only trained to predict streamflow, which is why it did not participate in the comparison of additional variables in this study. Note, however, that even though the model does not explicitly model additional physical states, it is nevertheless possible—to some degree—to extract such information from the internal states with the help of a small amount of additional data (Lees et al., 2021b; Kratzert et al., 2019a).



### 2.4.2 Locally calibrated models

The following models follow a local calibration strategy, i.e., models are trained for each of the 141 calibration stations individually. This leads to one calibrated model setup per basin. In order to simulate streamflow for basins that are not initially calibrated (spatial or spatio-temporal validation), the model needs a calibrated parameter set to be transferred to the uncalibrated basin. This parameter set is provided by a (calibrated) donor basin. All locally calibrated models in this study used the same donor basins for the validation basins as described in Sect. 2.5. The individual models will be briefly described in the following (one model per paragraph). More details for each model can be found in the supplementary material.

**LBRM-CC-lumped:** The Large Basin Runoff Model (LBRM) – described by Crowley II (1983) with recent modifications to incorporate the Clausius Clapeyron relationship, LBRM-CC, described by Gronewold et al. (2017) – is a lumped conceptual rainfall-runoff model developed by NOAA Great Lakes Environmental Research Laboratory (GLERL) for use in simulation of total runoff into the Great Lakes (e.g., used by U.S. Army Corps of Engineers in long term water balance monitoring and seasonal water level forecasting). The model will be referred to as "LBRM-CC-lumped" hereafter. Inputs to LBRM-CC-lumped include daily areally averaged (lumped) precipitation, maximum temperature, and minimum temperature, as well as contributing watershed area. All these inputs are based on the common dataset described in Sect. 2.2. For each basin, LBRM-CC-lumped's nine parameters were calibrated against the Kling-Gupta efficiency (KGE) (Gupta et al., 2009) using a Dynamically Dimensioned Search (DDS) algorithm (Tolson and Shoemaker, 2007) run for 300 iterations within the OSTRICH optimization package (Matott, 2017). A list of the calibrated nine parameters is given in the supplementary material (Tab. S3). LBRM-CC-lumped was modified for this project to output actual evapotranspiration (AET) in addition to the conventional outputs, which included snow water equivalent. LBRM does not provide a soil moisture as a volume per volume. Accordingly, we used the LBRM-CC-lumped "Upper Soil Zone Moisture" [mm] as a proxy for surface soil moisture. It should be noted that subsequent to the posting of model results, an error was found in the calibration setup that resulted in ignoring an important constraint on the proportionality constant controlling PET (see Lofgren and Rouhana, 2016, for an explanation of this constraint). In addition, the LBRM-CC-lumped modelling team found that an improved representation of the long term average temperatures applied in the PET formulation improves results in some tested watersheds (PET is a function of the difference between daily temperature and long term average temperature for the day of year). Accordingly, accounting for these PET modifications may improve the LBRM-CC-lumped performance in simulating AET in future studies. Because LBRM/LBRM-CC is conventionally used for simulating seasonal and long term changes in runoff contributions to the lakes' water balance, evaluation has previously focused on simulated runoff; simulated AET, SSM, and SWE are not typically evaluated. However, LBRM studies such as (Lofgren and Rouhana, 2016; Lofgren et al., 2011) did evaluate simulated AET leading to the development of the LBRM-CC-lumped. More model details are given in the supplementary material (Sect. S.2.2).

**HYMOD2-lumped:** HYMOD2 (Roy et al., 2017) is a simple conceptual hydrologic model simulating rainfall-runoff processes on a basin scale. Due to its lumped nature, it will be referred to as "HYMOD2-lumped" hereafter. It produces daily streamflow at basin level (lumped) by using daily precipitation and PET. The latter is calculated in this study using the Hargreaves method (Hargreaves and Samani, 1985) using the minimum and maximum daily temperatures. Additionally, the version





of HYMOD2-lumped used in this study includes snowmelt and rain-snow partitioning, for which temperature and relative
humidity data are required. The basin drainage area and latitude were the other two information relevant for the version of
the model implemented in this study. All forcing data and geophysical basin characteristics used are provided in the common
dataset (Sect. 2.2). In total, eleven model parameters are calibrated against the KGE using the Shuffle Complex Evolution
(SCE) algorithm (Duan et al., 1992). SCE was setup with two complexes and a maximum of 25 (internal) loops and the algo-
rithm converged on average after 1800 model evaluations. One calibration trial was performed for each basin. The list of the
eleven calibrated parameters is provided in the supplementary material (Tab. S4). The model variables for AET, and SWE are
vertically lumped and saved as model outputs to evaluate them against reference datasets. The proxy variable used for SSM is
the storage content in the nonlinear storage tank ($C$) from which the runoff is derived (see schematic diagram of HYMOD2
shown in Fig. 4 of Roy et al. (2017)). This storage content is written as a function of the height of water in the tank as follows,
$C(t) = C_{\max} \times (1 - (1 - (H(t)/H_{\max}))^{1+b})$. $H$ and $H_{\max}$ represent the height and maximum height of water, respectively,
while $b$ is a unitless shape parameter and $C_{\max}$ is the maximum storage capacity. Additional model details can be found in the
supplementary material (Sect. S.2.3).

    **GR4J-lumped, HMETS-lumped, Blended-lumped:** The models GR4J (Perrin et al., 2003), HMETS (Martel et al., 2017),
and the Blended model (Mai et al., 2020a) are setup within the Raven hydrologic modeling framework (Craig et al., 2020). The
models are setup in lumped mode and will therefore be called "GR4J-lumped", "HMETS-lumped", and "Blended-lumped",
respectively, hereafter. The three models run on a daily time setup. The forcings required are daily precipitation, as well as
minimum and maximum daily temperature. The models require the following static basin attributes which are all derived
based on the common set of geophysical data (Sect. 2.2): basin center latitude and longitude in [°], basin area in [km$^2$], basin
average elevation in [m], basin average slope in [°], and fraction of forest cover in basin (only HMETS-lumped and Blended-
lumped). The models are calibrated against KGE using the DDS algorithm. The models GR4J-lumped and HMETS-lumped
were given a budget of 500 model evaluations per calibration trial while the Blended-lumped model, that has more parame-
ters than the other two models, was given a budget of 2000 model evaluations per trial. In total, 50 independent calibration
trials per model and basin were performed; the best trial (largest KGE) was designated as the calibrated model. The tables of
calibrated parameters can be found in the supplementary material (Tab.s S5 to S7). For model evaluation, the model outputs
for actual evapotranspiration (AET) are saved using the Raven custom output `AET_Daily_Average_BySubbasin`. The
surface soil moisture SSM is derived using the saved custom output `SOIL[0]_Daily_Average_BySubbasin` which
is water content $\Theta$ in [mm] of the top soil layer and is converted using the unitless porosity $\phi$ and soil layer thickness $\tau_0$
in [mm] into the unitless soil moisture SSM $= \Theta/(\phi \cdot \tau_0)$). The snow water equivalent estimate is the custom Raven out-
put `SNOW_Daily_Average_BySubbasin`. Additional details for these three models can be found in the supplementary
material (Sect. S.2.4, S.2.5 and S.2.6).
**Blended-Raven:** The semi-distributed version of the above Blended model is also setup within the Raven hydrologic mod-
eling framework Raven. Setup is similar to the Blended-lumped model with the difference that instead of modelling each basin
in lumped mode, here, each basin is broken into subbasins (in total 2187 across all calibration basins and overall 2170 in vali-
dation basins) and these subbasins were further discretized into two different hydrological response units (lake and non-lake).



The simulated streamflows are routed between subbasins using the common routing product (Sect. 2.3) within Raven and is

hence called "Blended-Raven" hereafter. The model produces simulation results at the subbasin level but outputs are only saved for the most downstream subbasins (141 calibration, 71 validation). The model uses the same forcing variables (gridded and not lumped) and basin attributes as the Blended-lumped model. Calibration, validation and evaluation are exactly the same as described for the Blended-lumped model above. The model parameters calibrated can be found in the supplementary material (Sect. S.2.7 and Tab. S8).

**VIC-Raven:** The Variable Infiltration Capacity (VIC) model (Liang et al., 1994; Hamman et al., 2018) is a semi-distributed macroscale hydrological model that has been widely applied in water and energy balance studies (Newman et al., 2017; Xia et al., 2018; Yang et al., 2019). In this study, VIC was forced with the gridded RDRS-v2 meteorological data including precipitation, air temperature, atmospheric pressure, incoming shortwave radiation, incoming longwave radiation, vapor pressure, and wind speed at the 6 hour time step and 10 km by 10 km spatial resolution. Other data, such as topography, soil, and land

cover, are derived from those datasets as described in Sect. 2.2. VIC was used for runoff generation simulation at grid-cell level, and the resultant daily gridded runoff and recharge fluxes were aggregated to subbasin level for the Raven routing, which then produced daily streamflow at the outlet of each catchment. The VIC model using Raven routing will be referred to as "VIC-Raven" hereafter. In total, 14 parameters of the VIC-Raven model were calibrated at each of the 141 calibration catchments using the DDS algorithm to maximize the objective metric of the KGE. The optimization is repeated for 20 trials, each

with 1000 calibration iterations, and the best result out of 20 was designated as the calibrated model. The VIC model outputs including the soil moisture, evaporation, and snow water equivalent in snow pack are dumped for model evaluation. A list of the 14 calibrated model parameters and more details about the VIC-Raven setup can be found in the supplementary material (Tab. S9 and Sect. S.2.8, respectively).

### 2.4.3 Regionally calibrated models

Models following the regional calibration strategy focused on simultaneously calibrating model parameters to all calibration stations within a region (rather than estimating parameters for each gauged basin individually). All regionally calibrated models utilized the same six subdomains (here called "regions") which are the Ottawa River watershed and the local watersheds of Lake Erie, Lake Ontario, etc. (see Fig. 1C). This results in six model setups for the whole study domain for each of the regionally calibrated models. These models can be run for validation stations using those regional setups as soon as it is known

in which region the validation basin is located.

**SWAT-Raven:** The Soil and Water Assessment Tool (SWAT) was originally developed by the US Department of Agriculture and is a semi-distributed, physical-based hydrological and water quality model (Arnold et al., 1998). For each of the six regions, the geophysical dataset (DEM, soil, and land-use), as presented in Sect. 2.2, were used to create subbasins which were further discretized into hydrological response units. Subbasin averaged daily forcing data (maximum and minimum temperature as

well as precipitation), as presented in Sect. 2.2, were then used simulate land-surface processes. The vertical flux (total runoff) produced at subbasin spatial scale was then fed to a lake and river routing product (Han et al., 2020), integrated into Raven modelling framework (Craig et al., 2020), for subsequent routing. Note that the SWAT subbasins and routing product subbasins





are not equivalent and hence aggregation to routing product subbasins was necessary similar to the aggregation for gridded models. The integrated "SWAT-Raven" model was calibrated against daily streamflow using the OSTRICH platform (Matott,

2017) by optimizing median KGE values of all the calibration gauges in each region using 17 SWAT-related and two Raven related parameters. See the supplementary material (Sect. S.2.9 and Tab. S10) for further details of the parameters and model. The calibration budget consisted of 3 trials, each with 500 iterations and the best result out of the three trials was designated as the calibrated model. SWAT-simulated daily actual evapotranspiration, volumetric soil moisture (aggregated for the top 10 cm) and snow water equivalent at subbasin spatial scale were used to further evaluate the model.

**WATFLOOD-Raven:** WATFLOOD is a semi-distributed, gridded, physically-based hydrological model (Kouwen, 1988). A separate WATFLOOD model for each of the six regions (Fig. 1C) was developed using the common geophysical datasets as well as hourly precipitation and temperature data at a regular rotated grid of about 0.09 decimal degrees ($\approx$ 10 km). Model grids were further discretized into grouped response units. Hourly gridded runoff and recharge to lower zone storage (LZS) were aggregated to subbasin level for the Raven routing, which then produced hourly streamflow at the outlet of each catch-

ment. Streamflow calibration and validation of the integrated "WATFLOOD-Raven" was carried out using the same methodology (different parameters) as described in the previous section for SWAT-Raven. We refer to the supplementary material (Sect. S.2.10 and Tab. S11) for further details of the eleven WATFLOOD and six Raven parameters used during model calibration. Actual evapotranspiration and snow water equivalent are direct outputs of the WATFLOOD model and were used to evaluate the model. Volumetric soil moisture, however, is not a state variable of the model. Hence, the rescaled upper zone

storage (UZS) was used as a proxy to qualitatively evaluate the volumetric soil moisture.

**MESH-CLASS-Raven:** MESH (Modélisation Environnementale communautaire-Surface and Hydrology) is a community hydrologic modeling platform maintained by ECCC (Pietroniro et al., 2007). MESH uses a grid-based modelling system and accounts for sub-grid heterogeneity using the Grouped Response Units (GRU) concept from WATFLOOD (Kouwen et al., 1993). GRUs aggregate sub-grid areas by common attributes (e.g., soil and vegetation characteristics) and facilitate parameter

transferability in space (e.g. Haghnegahdar et al., 2015). The MESH model setup here is using the CLASS land-surface scheme to produce grid-based model outputs, which are passed to the Raven routing to generate the streamflow at the locations used in this study. The model will hence be referred to as "MESH-CLASS-Raven" hereafter. The following meteorologic forcing variables from the RDRS-v2 dataset (Sect. 2.2) are used to drive the CLASS land surface scheme: incoming shortwave radiation, incoming longwave radiation, precipitation rate, air temperature, wind speed, barometric pressure, and specific humidity

(all provided at or near surface level). The model simulates vertical energy and water fluxes for vegetation, soil, and snow. The MESH-CLASS-Raven grid is the same as the forcing data ($\approx$ 10 km by 10 km) and produces outputs at an hourly time step. The MESH-CLASS streamflow outputs were aggregated to a daily time step and then aggregated to subbasin level for the Raven routing, which then produced daily streamflow at the outlet of each subbasin. Streamflow calibration and validation of the integrated MESH-CLASS-Raven model was carried out using the same methodology (different parameters) as described

in the previous sections for SWAT-Raven and WATFLOOD-Raven. The calibration algorithm chosen is the DDS algorithm, with a single trial and a maximum of 240 iterations (per region). A list of the 20 model parameters calibrated is available in the supplementary material (Sect. S.2.11 and Tab. S12). The calibration was computationally time consuming (at least 2 weeks for





each region). For model evaluation, the MESH output control flag was used to activate the output of the daily average auxiliary variables (AET, SSM, SWE). The actual evapotranspiration (AET in [mm]) was directly computed by the model while the surface soil moisture (SSM in [m$^3$/m$^3$]) was obtained by adding the volumetric liquid water content of the soil (in [m$^3$/m$^3$]) to the volumetric frozen water content of the soil (in [m$^3$/m$^3$]). The snow water equivalent (SWE in [mm]) was obtained by adding the snow water equivalent of the snowpack at the surface (in [mm]) with liquid water storage held in the snow (e.g., meltwater temporarily held in the snowpack during melt, in [mm]).

**MESH-SVS-Raven:** The MESH (Modélisation Environnementale communautaire-Surface and Hydrology) model not only includes the CLASS Land-Surface Scheme (LSS) but also the Soil, Vegetation, and Snow (SVS; Alavi et al., 2016) LSS and the Watroute routing scheme (Kouwen, 2018). MESH allows, for example, to emulate the GEM-Hydro-Watroute model outside of ECCC informatics infrastructure. In this study however, the hourly MESH-SVS gridded outputs of total runoff and drainage were provided to the Raven routing scheme, which is computationally less expensive than the Watroute routing scheme (more info in supplementary material Sect. S.2.13). The model will be referred to as "MESH-SVS-Raven" hereafter. The following meteorological forcing variables from the RDRS-v2 dataset (Sect. 2.2) are used to drive the SVS LSS (same as used for MESH-CLASS-Raven): incoming shortwave radiation, incoming longwave radiation, precipitation rate, air temperature, wind speed, barometric pressure, and specific humidity. SVS also needs geophysical fields such as soil texture, vegetation cover, slope, and drainage density. All of these variables were derived from the common geophysical datasets used for this project (Sect. 2.2), except drainage density, which is derived from the NHN dataset over Canada (Natural Resources Canada, 2020) and from the NHD dataset over the United States of America (USGS, 2021). In this study, the MESH-SVS-Raven grid is the same as the forcing data (around 10 km by 10 km) and SVS was run using a 10 min time step. The Raven routing was used with a 6 hour time step. Routing results were aggregated to a daily time step for model performance evaluation. MESH-SVS-Raven was calibrated using a global calibration approach (Gaborit et al., 2015) but applied for each of the six regions of the complete area of interest, therefore resulting in what is called in this study a "regional" calibration method. The objective function consists of the median of the KGE values obtained for the calibration basins located in a region. The calibration algorithm consists of the DDS algorithm, with a single calibration trial and a maximum of 240 iterations, which took about 10 days to complete for each region. The list of the 23 parameters calibrated is provided in the supplementary material (Sect. S.2.12 and Tab. S13). Regarding the MESH-SVS-Raven auxiliary variables required for additional model evaluation, daily averages valid from 00 to 00 UTC were computed based on the hourly model outputs. For surface soil moisture (SSM), the average of the first two SVS soil layers was computed in order to obtain simulated soil moisture values valid over the first 10 cm of the soil column, which is the depth to which the GLEAM SSM values correspond. AET is a native output of the model, and the average SWE over a grid-cell was computed based on a weighted average of the two SVS snowpacks: SVS simulates snow over bare ground and low vegetation in parallel to snow under high vegetation (Alavi et al., 2016; Leonardini et al., 2021).

**GEM-Hydro-Watroute:** GEM-Hydro (Gaborit et al., 2017b) is a physically based, distributed hydrologic model developed at Environment and Climate Change Canada (ECCC). It relies on GEM-Surf (Bernier et al., 2011) to represent five different surface tiles (glaciers, water, ice over water, urban, and land). The land tile is represented with the SVS LSS (Alavi et al., 2016; Husain et al., 2016; Leonardini et al., 2021). GEM-Hydro also relies on WATROUTE (Kouwen, 2018), a 1D hydraulic model,


to perform 2D channel and reservoir routing. The model will be referred to as "GEM-Hydro-Watroute" in this study. It was preferred to use its native gridded routing scheme Watroute during this study, instead of the common Raven routing scheme used by other hydrologic models, in order to investigate the potential benefit of calibration for the experimental ECCC hydrologic forecasting system named the National Surface and River Prediction System (NSRPS; Durnford et al., 2021), which relies on GEM-Hydro-Watroute. GEM-Surf forcings are the same as those used for MESH-SVS-Raven, except that GEM-Surf needs both wind speed and wind direction (available in RDRS-v2), as well as surface water temperature and ice cover fraction (taken from the analyses of the ECCC Regional Deterministic Prediction System). Regarding the geophysical fields needed by GEM-Surf, they consist of the same as those used for MESH-SVS-Raven, except that some additional fields are required, like surface roughness and elevation (taken or derived from the common geophysical datasets used in this project). Moreover, the Watroute model used here mainly relies on HydroSHEDS 30 arcsec ($\approx$ 1 km) flow direction data (HydroSHEDS, 2021), GMTED 2010 elevation data (USGS, 2010), and on CCI-LC 2015 vegetation cover (ESA, 2015). The surface component of GEM-Hydro-Watroute was employed here with the same resolution as the forcings ($\approx$ 10 km), and with a 10 min time step, with outputs aggregated to hourly afterwards. The Watroute model used here has a 30 arcsec ($\approx$ 1 km) spatial resolution and a dynamic time step comprised between 30 and 3600 s. Daily flow averages were used when computing model performances. GEM-Hydro-Watroute is computationally intensive: it was not optimized to perform long-term simulations but is rather oriented towards large-scale forecasting; moreover, the Watroute routing scheme is computationally expensive, and its coupling with the surface component is not optimized either. See supplementary material Sect. S.2.13 for more information about GEM-Hydro-Watroute computational time. It is therefore very challenging to calibrate GEM-Hydro-Watroute directly. Several approaches have been tried to circumvent this issue in the past, each having its own drawbacks (see Gaborit et al., 2017b; Mai et al., 2021). In this study, a new approach is explored, consisting of calibrating some SVS and routing parameters with MESH-SVS-Raven, transferring them back in GEM-Hydro-Watroute (more info in supplementary material Sect. S.2.13 and Tab. S14), and further manually tuning Watroute Manning coefficients afterwards. However, because this approach leads to different performances for MESH-SVS-Raven and GEM-Hydro-Watroute for several reasons explained in the supplementary material Sect. S.2.13, they are presented here as two different models. The exact same approach as for MESH-SVS-Raven was used for GEM-Hydro-Watroute streamflow validation and for the additional model evaluation variables.

## 2.5 Model calibration and validation setup and datasets

The models participating in this study were calibrated and validated based on streamflow observations made available by either Water Survey Canada (WSC) or the U.S. Geological Survey (USGS). The streamflow gauging stations selected for this project had to have at least a (reported) drainage area of 200 km$^2$ (to avoid too flashy watershed responses in the hydrographs) and less than 5% of missing (daily) data between 2000 and 2017 which is the study period, i.e. warmup period (January to December 2000), calibration period (January 2001 to December 2010) and validation period (January 2011 to December 2017). The low portion of missing data was to avoid not having enough data to properly estimate calibration and validation performance of the models.





Besides a minimum size and data availability, the streamflow gauges need to be either downstream of a low-human impact watershed (objective 1) or most downstream of areas draining into one of the five Great Lakes or into the Ottawa River (objective 2). Objective 1 was defined to give all models – especially the ones without the possibility to account for watershed management rules – to perform well. Objective 2 was chosen since the ultimate goal of most models is to estimate the flow
into the lakes (or the Ottawa River).

The stations were selected such that they are distributed equally across the six regions (five lakes and Ottawa river) and would be used for either calibration (cal) or validation (val). The assignment of a basin to be either used for calibration or validation was done such that two-third of the basins (141) are for training and the remaining basins (71) are for validation. There are 33 stations located within the Lake Superior watershed (21 cal, 12 val), 45 within the Lake Huron watershed (29 cal,
16 val), 35 stations within the Lake Michigan watershed (27 cal, 8 val), 48 locations within the Lake Erie watershed (36 cal, 12 val), 31 stations within the Lake Ontario watershed (21 cal, 10 val) and 20 stations located within the Ottawa River watershed (7 cal, 13 val). The location of these 212 streamflow gauging stations is displayed in Fig. 1. A list of the 212 gauging stations including their area, region they are located in, objectives they were selected for, and whether they were used for calibration or validation is given in the Supplementary Material (Tab. S15).

To summarize the steps of calibration and validation of all models regarding streamflow:

A. *calibration* performed for 141 calibration locations for period January 2001 to December 2010 (2000 discarded as warmup)

B. *temporal validation* performed for 141 locations where models were initially calibrated but now evaluating the simulated streamflow from January 2011 until December 2017

C. *spatial validation* performed for 71 validation locations from January 2001 until December 2010

D. *spatio-temporal validation* performed for 71 validation locations from January 2011 until December 2017

The temporal validation (B; basin known, time period not trained) can be regarded to be the easiest of the three calibration tasks (assuming climate change impacts are not too strong, no major landcover changes happened, etc.). The spatial validation (C; time period trained but location untrained) can be regarded to be more difficult especially for locally calibrated models
given that one either needs a global/ regional model setup or a good parameter transfer strategy for ungauged/ uncalibrated locations. The spatio-temporal validation (D) can be regarded to be the most difficult validation experiment as both location and time-period were not included in model training.

All locally calibrated models used the same (simple) donor basin mapping for assigning model parameter values from calibrated basins to other basins used for spatial and spatio-temporal validation. The donor basins were assigned using the
following strategy: (a) if there exists a nested basin (either upstream or downstream) that was trained use that basin as donor, (b) if there are multiple of those nested candidates use the one that is the most similar with respect to drainage area, and (c) if there is no nested basin trained, use the calibration basin which spatial centroid is the closest to the spatial centroid of the validation basin. There are certainly more appropriate/advanced ways to do regionalization, but the main objective was



to ensure that the models used the same approach. We wanted to avoid favouring a few models by testing more advanced
approaches with them and then applying that method out to other models which might have favoured other strategies. The
donor basin mapping is provided in the Supplementary Material (Tab. S16).

The WSC and USGS streamflow data were downloaded by the project leads, units converted to be consistently in $[m^3/s]$,
and then shared with the team in both CSV as well as NetCDF format to make sure all team members used the same version
of data for calibration. Note that at first only data of the 141 calibration locations was shared. The validation locations and
respective observations were only made available after the calibration of all models was finished to allow for a true blind
(spatial and spatio-temporal) validation.

## 2.6 Model evaluation setup and datasets

In order to assess the model performance beyond streamflow, a so-called model evaluation was performed. Model evaluation
means that a model was not (explicitly) trained against any of the data used for evaluation. We distinguish model evaluation
from model validation of streamflow because when building models, although modelling teams knew their model would be
assessed based on streamflow validation, they did not know in advance that additional variables beyond streamflow would be
assessed against other observations (this was only decided late in the study). It is not known if models have been trained in
previous studies against any of such data and, for example, model structures or process formulations might have been informed
by such a preliminary training. Such kind of model developments are however the nature of modeling in general and assumed
to exactly determine the quality of a model that is to be assessed here.

The evaluation datasets were chosen based on the following criteria: (a) they had to be gridded to allow for a consistent
evaluation approach across all additional variables, (b) the time step of the variables had to be preferably daily to match the
time step of the primary streamflow variable, (c) they had to be available over the Great Lakes for a long time period between
2001 and 2017 (study period) to allow for a reliably long record in time and space, (d) they had to be open-source to allow for
reproducibility of the analysis, (e) they had to be variables typically simulated in a wide range of hydrologic and land-surface
models, and (f) they had to have been at least partially derived on distributed observational datasets for the region (e.g., ground
or satellite observations).

Two evaluation variables were taken from the GLEAM v3.5b dataset (Martens et al., 2017). GLEAM is set of algorithms us-
ing satellite observations of climatic and environmental variables in order to estimate different components of land evaporation
as well as surface and root-zone soil moisture, potential evaporation, and evaporative stress conditions. The first variable used
in this project is actual evapotranspiration (AET; variable 'E' in dataset; see Fig. 2B) given in $[mm/d]$ and the second variable
is surface soil moisture (SSM; variable 'SMsurf' in dataset; see Fig. 2C) given in $[m^3/m^3]$ and valid for the first 10 cm of the
soil profile. Both variables are available over the entire study domain on a 0.25° regular grid and for the time period from 2003
until 2017 with daily resolution.

The third evaluation variable is snow water equivalent (SWE) available in the ERA5-Land dataset (Muñoz Sabater, 2019)
(variable 'sd' therein; see Fig. 2D). ERA5-Land combines a large amount of historical observations into global estimates of
surface variables using advanced modelling and data assimilation systems. ERA5-Land is available for the entire study period





and domain on a daily temporal scale and a $0.1°$ regular grid. The data are given in $[m$ water equivalent$]$ and have been converted to $[kg/m^2]=[mm]$ to allow comparison with model outputs.

The ERA5-Land dataset was chosen over available ground-truth SWE observations like the CanSWE dataset (Vionnet et al., 2021) because (a) the ERA5-Land dataset is available over the entire modeling domain while CanSWE is only available for the Canadian portion, (b) the ERA5-Land dataset is gridded allowing for similar comparison approaches as used for AET and SSM, (c) the ERA5-Land dataset is available on a daily scale while the frequency of data in CanSWE varies from biweekly to monthly which would have limited the number of data available for model evaluation. We however provide a comparison of the

ERA5-Land and CanSWE observations by comparing the grid cells containing at least one snow observation station available in CanSWE and derive the KGE of those two time series over the days both datasets provide estimates. The results show that 83% of the locations show an at least medium agreement between the ERA5-Land SWE estimates and the SWE observations provided through the CanSWE dataset (KGE larger than or equal 0.48). A detailed description and display of this comparative analysis can be found in the Supplementary Material (Sect. S.1).

These three auxiliary evaluation datasets were downloaded by the project leads, merged into single files from annual files such that the team only had to deal with one file per variable. The data were subsequently cropped to the modeling domain and then distributed to the project teams in standard NetCDF files with a common structure, e.g. spatial variables are called the same way, to enable an easier handling in processing. Besides these pre-processing steps, the gridded datasets were additionally aggregated to the basin-scale resulting in 212 files – one for each basin – to allow for not only a grid-cell-wise comparison

but also a basin-level comparison. The reader should refer to Sect.s 2.7 and 2.8 for more details on performance metrics and basin-wise/grid-cell-wise comparison details, respectively.

## 2.7  Performance metrics

In this study the Kling-Gupta efficiency (KGE) (Gupta et al., 2009) and its three components $\alpha$, $\beta$, and $r$ (Gupta et al., 2009) are used for streamflow calibration and validation as well as model evaluation regarding AET, SSM, and SWE. The component $\alpha$

measures the relative variability of a simulated versus an observed time series (e.g., $Q(t)$ or $\mathrm{AET}(x',y',t)$ or $\mathrm{SWE}(x',y',t)$ of one grid cell $(x',y')$). The component $\beta$ measures the bias of a simulated versus an observed time series while the component $r$ measures the Pearson correlation of a simulated versus an observed time series. The overall KGE is then based on the Euclidean distance from its ideal point in the untransformed criteria space and converting the metric to the range of the Nash-Sutcliffe efficiency, i.e. optimal performance results in a KGE and NSE of 1 and suboptimal behavior is lower than 1:

$$\mathrm{KGE} \;=\; 1-\sqrt{(1-\alpha)^2+(1-\beta)^2+(1-r)^2} \;\; \in (-\infty,1] \tag{1}$$

To make comparisons easier, we transformed the range of the KGE components to match the range of the KGE. We introduce the following component metrics:

$$\mathrm{KGE}_\alpha \;=\; 1-\sqrt{(1-\alpha)^2}=1-\sqrt{(1-\sigma_{y_{\mathrm{sim}}}/\sigma_{y_{\mathrm{obs}}})^2} \;\; \in (-\infty,1] \tag{2}$$

$$\mathrm{KGE}_\beta \;=\; 1-\sqrt{(1-\beta)^2}=1-\sqrt{(1-\overline{y_{\mathrm{sim}}}/\overline{y_{\mathrm{obs}}})^2} \;\; \in (-\infty,1] \tag{3}$$

$$\mathrm{KGE}_r \;=\; 1-\sqrt{(1-r)^2} \;\; \in (-\infty,1] \tag{4}$$





**Table 2. Performance levels.** The table shows the ranges per metrics to qualify as excellent, goo, medium, or poor performance. The KGE component thresholds (Eq.s 2 to 4) are defined independently while the overall KGE is the result of the thresholds combined through Eq. 1.

|  | poor performance | medium performance | good performance | excellent performance |
|---|---|---|---|---|
| $KGE_\alpha$ | $(-\infty, 0.70)$ | $[0.70, 0.80)$ | $[0.80, 0.90)$ | $[0.90, 1.0]$ |
| $KGE_\beta$ | $(-\infty, 0.70)$ | $[0.70, 0.80)$ | $[0.80, 0.90)$ | $[0.90, 1.0]$ |
| $KGE_r$ | $(-\infty, 0.70)$ | $[0.70, 0.80)$ | $[0.80, 0.90)$ | $[0.90, 1.0]$ |
| KGE | $(-\infty, 0.48)$ | $[0.48, 0.65)$ | $[0.65, 0.83)$ | $[0.83, 1.0]$ |

with $y$ being a time series (either simulated $sim$ or observed $obs$) and $r$ being the Pearson correlation coefficient.

We use the overall KGE to estimate the model performances regarding streamflow (components are shown in the Supplementary Material only), AET, and SWE. For surface soil moisture we used the correlation component of KGE ($KGE_r$) since most models do not explicitly simulate soil moisture but only saturation or soil water storage in a conceptual soil layer where depth is not explicitly modelled. We therefore agreed on reporting only standardized soil moisture simulations defined as

$$SSM' \;=\; \frac{SSM - \mu_{SSM}}{\sigma_{SSM}} \tag{5}$$

where $\mu_{SSM}$ is the mean and $\sigma_{SSM}$ denotes the standard deviation of a surface soil moisture time series. The mean and variance of those standardized time series $SSM'$ hence collapse to 0 and 1, respectively, and the KGE components $\beta$ and $\alpha$ are noninformative. The Pearson correlation $r$ however is informative. Note that the Pearson correlation for standardized and non-standardized datasets are the same ($r(SSM') = r(SSM)$).

Several levels of model performance are introduced in this study to allow for a categorization of results and models. The thresholds defined in the following for the individual components have been chosen based on expert opinions and reflect the subjective level of quality expected from the models. The levels can be adapted depending on the stronger or looser expectations. The thresholds of the components are then combined to the (derived) thresholds for KGE. This approach to defining performance level bins was also used by Mei et al. (2022).

We defined a "medium" performance to not over- or underestimate bias and variability by more than 30%. A correlation is expected to be at least 0.7. All performances below this are regarded to be of "poor" quality. A "good" performance is defined as bias and variability to be over- or underestimated at most by 20% and a correlation to be at least 0.8. An "excellent" performance is defined to be not more than 10% over- or underestimating the variability or bias and a correlation of at least 0.9. The resulting ranges for the KGE and its three component performances are derived using Eq. 1 to Eq. 4 and summarized in Tab. 2. These performance levels are consistently used throughout this study to categorize the model performances regarding streamflow and additional variables.





## 2.8  Basin-wise and grid-wise model comparisons for individual simulated variables

We decided to use two approaches to compare model outputs with streamflow observations and the reference datasets for AET,
SSM, and SWE in order to account for the wide range of spatial discretization approaches used by the participating models. Six
models were setup in a lumped mode (basin-wise output), two models operated on a subbasin discretization, and five models
are setup in gridded mode where the model grids do not necessarily match the grid of the AET, SSM, or SWE reference
datasets. Please refer to Tab. 1 for the spatial resolution of the individual models.

In the calibration and validation phases of the project (Sect. 2.5), the simulated time series of streamflow are compared in the
traditional way with observed time series of streamflow at gauging stations. We used the Kling-Gupta efficiency metric (Eq. 1)
to compare the simulated daily streamflow estimates with the corresponding observations.

In the evaluation phase (Sect. 2.6) the additional variables were compared to their simulated counterparts. The additional
variables chosen for evaluation are gridded and on a daily time scale. Model evaluation for additional variables are compared
following two strategies. The first comparison is performed basin-wise meaning that reference datasets and simulations are
both aggregated to basin-level, e.g. gridded evapotranspiration $\mathrm{AET}(x,y,t)$ is aggregated to area-weighted mean evapotran-
spiration $\overline{\mathrm{AET}}(t)$ of all grid cells that contribute to a basin domain. This results into 212 observed time series per variable
and 212 simulated time series per model and variable. The corresponding basin-wise observed and simulated time series are
then compared using KGE (Eq. 1) for evapotranspiration and snow water equivalent, and the Pearson correlation coefficient
(Eq. 4) for surface soil moisture. The second comparison strategy is to compare all models grid-cell-wise thereby appropri-
ately enabling the evaluation of how well distributed models can simulate spatial patterns within basins. In this approach all
model outputs are regridded to the grid of the reference datasets (i.e., 0.25° regular grid for GLEAM's evapotranspiration and
surface soil moisture and 0.1° regular grid for ERA5-Land's snow water equivalent). 'Regridding' is here meant as the areally
weighted model outputs contributing to each grid cell available in the reference datasets; no higher-order interpolation was
applied. The reference (gridded) and simulated (regridded) time series are then compared for each grid cell leading to a map of
model performance per variable (i.e., $\mathrm{KGE}(x,y)$). Again, the KGE is used for evapotranspiration and snow water equivalent,
and the Pearson correlation coefficient for surface soil moisture.

Note that also lumped models and models that are based on a subbasin discretization are "regridded" to the reference grids. In
those cases, each grid cell is overlay with the (sub)basin geometries and the area-weighted contribution of the (sub)basin-wise
outputs to the grid cell are determined.

Basin-wise comparisons match a lumped modelling perspective while grid-cell-wise comparisons are a better approach for
distributed models. The authors are not aware of past studies including grid-cell-wise comparisons across multiple models. This
may be due to regridding being a tedious task – especially considering a wide range of model internal spatial discretizations.
The regridding was performed by the project leads to ensure a consistent approach across all models. The team members
provided the project leads with their model outputs using the native model grid in a standardized NetCDF format that the group
had agreed on.





## 2.9 Multi-objective multi-variable model analysis

Besides analysing model performances individually for the variables of streamflow, actual evapotranspiration (AET), surface soil moisture (SSM), and snow water equivalent (SWE), a multi-objective analysis considering the performance of all variables at the same time was performed. Considering performance metrics for these four variables are available, each of these can be considered an objective and modellers would like to see all objectives optimized. From a multi-objective perspective model $A$ can only be identified to be truly superior compared to model $B$ when it is superior in at least one objective with all other $N-1$ objectives being at least equal to model $B$. In this case model $A$ is said to dominate model $B$ and model $B$ is dominated by model $A$. When multiple models $A$, $B$, $C$, ..., $M$ are under consideration, model $A$ is dominated when there is at least one other model that dominates model $A$ otherwise model $A$ is non-dominated (e.g., no model is objectively superior to model $A$). The set of non-dominated models forms the so-called Pareto front which is of unique interest as it defines the set of models where no objective decision can be made about which model is better than the others. Theoretically, any number between 1 to $M$ models can form the Pareto front.

The Pareto analysis, i.e. computing the set of models on the Pareto front, is carried out for each of the 212 basins. The $N=4$ objectives are KGE of basin-wise AET and SWE, as well as $\mathrm{KGE}_r$ of SSM derived for the entire period from 2001 to 2017 and the basin-wise KGE regarding streamflow derived for the validation period from 2011 to 2017. The validation period was chosen for streamflow to allow for a fair comparison with the evaluation variables as those have never been used for model training. Including the calibration period (2001 to 2010) for streamflow would have implicitly favoured models that are easier to calibrate.

In this analysis, the score for each model is defined by the number of times a model is part of the Pareto front (i.e., not dominated by another model). A perfect score is achieved if a model is a member of the non-dominated set of models in 212 of the 212 Pareto analyses (100%); the lower limit of this score is 0 (0%).

In order to shed some light on reasons for high/low scores of individual models, the Pareto analysis was repeated with only $N=3$ metrics at a time. Streamflow as a primary hydrologic response variable was included each time while one of the three evaluation variables was left out. This results in three additional scores for each model besides the score that is derived using all $N=4$ variables.

## 2.10 Interactive website

During the two years of this project an interactive website was developed as an interactive tool to explore study results and data beyond what can be shown in individual static plots. It is, for example, challenging to show the 212 times 13 hydrographs for the calibration and validation periods or to compare individual events at a specific location between a subset of models.

The website (Mai, 2022) is hosting all results for basin-wise model outputs of streamflow, AET, SSM, and SWE. The basin-specific time series can be displayed and specific event analysed by zooming into the time series plots.





Additional basin characteristics such as mean precipitation or temperature as well as slope or basin elevation can be shown through the drop-down menu to color markers on website. The user also has the option to choose "Ranking of models" in this coloring dropdown menu.

The website was used during the project to easily communicate results with team members and allow all modelling teams to explore model differences and features. We hope that the website helps with user engagement since all data and model outputs can be displayed without the necessity of downloading, processing, and plotting results of interest. While we see the potential to expand the characteristics and data displayed on this website, we focused on displaying only data produced and used in this study. Additional models might be added at a later point to this website. A requirement is that the models are setup and run

with the common dataset (Sect. 2.2), and the models are calibrated, validated and evaluated at the required locations and for all four variables. The website will incorporate updates to datasets (for example resulting from improved calibration or bug fixes) as they become available.

## 3   Results and Discussion

The results of four different comparative analyses performed in this study are presented in this section. The discussion of

each analysis will be presented right after the results to improve readability. Section 3.1 will present results and insights gained through the analysis of model performances regarding streamflow. The performances are compared for all thirteen participating models and the four experiments listed in Sect. 2.5, i.e., calibration, temporal validation, spatial validation, and spatio-temporal validation. Section 3.2 will focus on the outcomes of the analysis of the models' performance regarding the three additional variables (actual evapotranspiration, surface soil moisture, and snow water equivalent) when the outputs of each model are

lumped to basin-scale while Section 3.3 will analyse the performance at the grid-cell-scale. Section 3.4 will present the results of the multi-objective multi-variable model analysis in order to integrate the model performance across the four variables into one overall metric per model to allow for a high-level model intercomparison. The sections 3.2 to 3.4 include analyses of model performances regarding the auxiliary variables. Those will hence be based on only twelve models since the Machine Learning-based LSTM-lumped model is not trained to output such intermediate variables and would need fitting with additional data to

predict them.

### 3.1   Model performance regarding streamflow

The results of the streamflow performances across all models and location are presented in Fig. 3. The figure contains one panel for each of the experiments performed to compare the models. Each panel will be described and discussed in one of the following subsections, i.e. Sect. 3.1.1 describes calibration results, Sect. 3.1.2 describes the results of the temporal validation,

Sec .3.1.3 spatial validation, and Sect. 3.1.4 spatio-temporal validation. Results are summarized in Sect. 3.1.5.



**Figure 3. Model performance regarding streamflow.** The performance is shown for (A) the 141 calibration stations and (B) the 71 validation locations for the calibration period. The results for the validation period of the calibration and validation sites are shown in panels (C) and (D), respectively. In summary, panel (B) shows spatial validation, panel (C) shows temporal validation, and panel (D) shows spatio-temporal validation across the locations (x-axis) and the 13 models (y-axis). The locations are grouped according to their location within the six watersheds (vertical black lines) of Lake Erie (ERI), Lake Huron (HUR), Lake Michigan (MIC), Lake Ontario (ONT), Ottawa River (OTT), and Lake Superior (SUP). The horizontal black lines separate the machine-learning based global LSTM model from the models that are calibrated locally and the models that are calibrated per region. The performance is quantified using the Kling-Gupta efficiency (KGE). The median KGE performance of each model for each of the four evaluation scenarios is added as labels to each panel. The thresholds for medium, good, and excellent performance classifications are added as labels to the colorbar. Figures showing the results of the three components of KGE (bias, variability, correlation) can be found in the Supplementary Material (Figures S2 to S4). For the spatial distribution of these results as well as the simulated and observed hydrographs please refer to the website (www.hydrohub.org/grip-gl/maps_streamflow.html).





### 3.1.1 Experiment A: Calibration

First, all modeling groups were asked to calibrate their models regarding streamflow at 141 streamflow gauging stations during the time period starting January 2001 until December 2010. The forcings for the year 2000 were provided but assigned as warmup period and not included. It was known apriori that the models would be assessed by KGE and the median KGE across

all stations. The results are shown in Fig. 3A. Therein, the models (y-axis) are grouped according to the three types of calibration schemes (globally, locally, or regionally calibrated) while the gauges (x-axis) are grouped according to the lakes/river they ultimately drain into. The only globally calibrated model, i.e. LSTM-lumped, has almost perfect scores for all gauges. This is not surprising as this is training results that could theoretically be trained to achieve a KGE of 1.0 everywhere with ML models. Due to this, these results are usually not even shown in the ML community. They are added here for completeness. The locally

calibrated group of models, i.e, LBRM-CC-lumped, HYMOD2-lumped, GR4J-lumped, HMETS-lumped, Blended-lumped, Blended-Raven, and VIC-Raven, shows a median KGE performances between 0.80 (VIC-Raven) and 0.88 (Blended-lumped). Interestingly, there is no indication that semi-distributed locally calibrated models perform better (or worse) than the lumped models as, for example, the semi-distributed VIC-Raven model is the one with the lowest performance (median KGE of 0.80) while the semi-distributed Blended-Raven is one of the best (median KGE of 0.86). Nevertheless, all locally calibrated models

exhibit good or excellent performance levels (see Tab. 2) and we do not want to read too much into minor differences in KGE performance. There are no major differences between the six drainage regions for these models and no location can be identified to appears to have bad KGE values across all locally calibrated models which usually indicates data issues for this streamflow gauge rather than a model issue. The regionally calibrated models, i.e., SWAT-Raven, WATFLOOD-Raven, MESH-CLASS-Raven, MESH-SVS-Raven, and GEM-Hydro-Watroute show an expectedly lower performance than the locally calibrated

group of models. The KGE performances spread between 0.50 and 0.63 and can hence be classified to be of medium quality (see Tab. 2). The main reason why GEM-Hydro-Watroute has a significantly lower performance than MESH-SVS-Raven, despite the former mainly relying on parameters calibrated with the latter, is expected to be due to a bug that was present in the MESH-SVS-Raven model and related to the reading of vegetation cover from the geophysical files provided to the model. Note that this bug was not present in previous studies (Gaborit et al., 2017b; Mai et al., 2021) as it was due to the specific

NetCDF format used with MESH-SVS-Raven input/output files during this work. This led to the SVS LSS not using the right information for vegetation cover during calibration, and therefore to calibrated parameters that were not optimal for SVS inside GEM-Hydro-Watroute, where the reading of vegetation cover was done properly. Because discovered after this study was completed, there was no time left to restart the work for MESH-SVS-Raven and GEM-Hydro-Watroute. However, new results for both models could be added to the website in the near future. Some basins seem to be difficult to model across all

the regionally calibrated models. These basins are watersheds (a) with strong flow regulation (e.g., 02BD002, 02DD010), (b) that are located in urban areas (e.g., 02HA006, 02HA007, 02HC049, 04166500, 04174500), and (c) in agricultural regions in the US (e.g., 04085200, 04072150). Such basins appear in all regions except the Ottawa region (OTT) which is an indicator that the basins selected for calibration in the Ottawa River watershed might be neither strongly regulated nor in urban areas. Hence the set of basins might not be very representative when models are validated in this region.



### 3.1.2 Experiment B: Temporal validation

After calibration we performed three validation experiments. The first and easiest validation test was temporal validation using the same basins as in calibration but evaluating the performance for the time period from January 2011 to December 2017 (Fig. 3C). As expected the KGE performance is decreasing for all models. The median KGE of the Machine Learning model decreases by 0.15 but the model still maintains a median KGE of 0.82 which is the best overall median KGE of all models in the temporal validation experiment. The performance of the locally calibrated models are all very similar among each other with median KGE values between 0.74 and 0.79. The two Blended model setups (Blended-lumped and Blended-Raven) lose the most skill compared to calibration but are still the two best performing models in this group together with HYMOD2-lumped (median KGE of 0.79, 0.76, and 0.76, respectively). The differences in performance across the regionally calibrated models are larger: The best is WATFLOOD-Raven (median KGE of 0.62; decreased by 0.01 compared to calibration) while MESH-CLASS-Raven and GEM-Hydro-Watroute are showing the overall lowest median KGE values (median KGE of 0.45 and 0.46, respectively). Overall, the pattern of basins with low performance and good performance are similar to the patterns seen in calibration (see collocation of red and blue cells in Fig.s 3A and 3C) indicating that the performance decrease in temporal validation is similar for all stations and no station deteriorates significantly when the time period is changing.

### 3.1.3 Experiment C: Spatial validation

The second, more challenging validation experiment was to run models in the calibrated time period but at locations that were not used (or even known) during calibration (Fig. 3B). The ML-based model loses more of its overall performance than in temporal validation (decrease by 0.21 vs. 0.15) but is still significantly better than all other models (median KGE of 0.76 vs. second-best (Blended-lumped) model with median KGE of 0.61). The locally calibrated models show similar (medium) performances among each other with some gauges performing consistently poor (entire column is red) which can be attributed to the fact that all models used the same – probably not the most suitable – donor basin to transfer the parameters to the basin in question in order to derive a streamflow estimate. It is expected that the overall performance of the locally calibrated models could be improved by a more sophisticated approach determining the donor basins. It is unlikely that the poor performance of some locations is a model-specific problem as the locally calibrated models are diverse in setup and complexity (e.g., compare conceptual GR4J-lumped and semi-distributed, physically-based VIC-Raven). It should be mentioned that the two regions with large sets of calibrated basins and fewer basins in validation (e.g., Lake Michigan (MIC) and Lake Erie (ERI) watershed) show much better results and fewer stations with poor performance across all models. This is very likely due to the fact that the pool the donor basin can be chosen from is very large and the likelihood to assign a similar basin is higher. In contrast, the region of the Ottawa River basin had only a limited set of basins for calibration which was previously discussed to be potentially not very representative as the calibration set did not contain highly managed basins or basins in urban or agricultural areas. Besides that, the set of validation basins are all wide spread in space and in many cases the only available donor basin is far away and thus unlikely to be a good proxy for the validation location. The regionally calibrated models do not show as strong a loss of performance when validated spatially. The median KGE decreases on average by 0.16 (0.10 if outlier KGE of



MESH-CLASS-Raven is ignored) while the median KGE of locally calibrated models is decreasing by 0.27 (spatial validation performance compared to the respective median performance during calibration). The overall performance of the regionally calibrated models (median KGE between 0.47 and 0.52 while ignoring outlier median KGE of 0.17 for MESH-CLASS-Raven) is much closer to the performance of the locally calibrated models (median KGEs between 0.52 and 0.61) in spatial validation compared to the larger performance gap in calibration.

### 3.1.4 Experiment D: Spatio-temporal validation

The third validation experiment is the spatio-temporal validation (Fig. 3D) where the models are assessed for (a) basins that are not calibrated and (b) a time period that was not used for calibration. The spatio-temporal validation is a combination of the two previous (temporal and spatial) validation experiments and is hence regarded to be the most challenging validation experiment the models undergo in this study. The spatio-temporal validation shows consistent results with the two previous experiments. Almost all models degrade in performance compared to the spatial validation but not as much as the skill was reduced when comparing temporal validation with spatial validation. Some models even slightly increase their median performance (by at most 0.03) compared to the spatial validation (i.e., GR4J-lumped, WATFLOOD-Raven, MESH-CLASS-Raven, MESH-SVS-Raven) but we do not want to read too much into this as the skill of those models was already poor to begin with. Again, the performance of the regionally calibrated models is much closer to the average performance of the locally calibrated models as compared to the large performance gaps in calibration and temporal validation. As expected, results generally show temporal validation to be easier task than spatial (or spatio-temporal) validation. The overall performance of locally calibrated models is expected to improve when a more sophisticated method to determine donor basins is applied.

### 3.1.5 Summary of streamflow performance experiments

In summary, the Machine Learning-based LSTM-lumped model is by far the best model not only in calibration but also in every validation scenario. The model skill is excellent in calibration, spatial and temporal validation and can still be classified as good in spatio-temporal validation. The highest ranking spatial and spatio-temporal validation performance levels of the LSTM model is noteworthy given their model training involved fitting roughly 300,000 model parameters. LSTM training procedures were therefore incredibly and, to many co-authors, surprisingly successful at avoiding overfitting.

The LSTM-lumped model shows significant improvements (especially in validation) compared to the preceding LSTM model in the GRIP-E project (Mai et al., 2021, called ML-LSTM therein). This is due to several reasons: (a) More training data (streamflow locations and forcings) were available in the GRIP-GL project which is a known ingredient to guarantee good performance of data-driven models, (b) the team setting up the model gained expertise, and (c) additional basin characteristics were used to train the models (Tab. S2). Besides the excellent skill of the model throughout, another advantage is its global setup, which means that it is easy to obtain streamflow at stations beyond the ones used here for validation. This strong performance of the LSTM-lumped model is in line with results from prior studies (Kratzert et al., 2019b).

The locally calibrated models – mostly conceptual and fast – might be good starting point for streamflow estimates – especially with a more advanced donor-basin mapping. Unless sophisticated parameter transfer methods are tested and employed,



locally calibrated models are not well suited for simulations in ungauged areas, due to their lack of spatial robustness. The impact of more sophisticated donor-basin mapping methods will be evaluated in a follow-up study. The performance level of the locally calibrated models can be classified as excellent to good in calibration, good in temporal validation, and medium in spatial and spatio-temporal validation. Even though the models are (strictly speaking) not the same as in the previous GRIP-E project, similar trends were found there as well. In GRIP-E, the calibration performances in calibration were good (median KGE of 0.63 to 0.78 for most models excluding SWAT) and significantly decreased in temporal validation which was the only validation performed in GRIP-E (median KGE values between 0.44 to 0.68 excluding SWAT).

The regionally calibrated models are more stable – meaning that they lose less performance relatively – compared to locally calibrated models and have the advantage that they provide setups over entire regions. This makes it straight forward to derive streamflow estimates for ungauged/untrained locations (spatial and spatio-temporal validation). The regionally calibrated models show medium performance levels in calibration, medium performances in temporal validation, and medium (on the edge to poor) performances in spatial and spatio-temporal validation with the exception of MESH-CLASS-Raven which is poor in both. MESH-SVS-Raven and WATFLOOD-Raven are performing very well compared to the other regionally calibrated models. The latter model is notable as this was the model that showed the weakest validation performance during GRIP-E. The regionally calibrated models generally have their lowest performances in managed, urban, and agricultural basins. This might be an indication that all these models could be improved (a) if associated processes like reservoir operation rules, tile drainage, or handling of processes over impervious/sealed areas are better represented in those models and (b) if models would specifically be trained for such basins since the representation of only 2-3 basins with those specific characteristics within the entire set of calibration basins of a region might not be sufficient to constrain/calibrate all parameters required for those processes. In other words, the models that were regionally calibrated during this work following a geographical subdivision of the whole domain into six regions may instead benefit from being calibrated over all agricultural, urban, and natural basins of the whole domain, following a morphological instead of a geographical subdivision.

Regarding possibility (a), a strong improvement of GEM-Hydro-Watroute was for example noticed when using the Dynamically Zoned Target Release (DZTR) reservoir model (Yassin et al., 2019) to represent regulated reservoirs in Watroute, for gauges 02KF005 (Ottawa River, nine upstream managed reservoirs), 02DD010 (Lake Nipissing) and 02AD012 (Lake Nipigon). In this study, GEM-Hydro-Watroute was the only model to include a reservoir regulation model in its routing scheme, for basins whose gauges are 02KF005, 02DD010, and 02LB005 and probably benefited from it for these gauges. However, given this unique strategy was only applied in 3 of the 212 studied basins, it had little impact on overall GEM-Hydro-Watroute comparative performance.

Regarding improvement possibility (b), a significant loss of performance can be noticed for the Ottawa region even with regionally calibrated models, regarding spatial validation. In this region, only a few calibration basins were selected and these generally include agricultural areas and were therefore not representative of the general morphology of the region, which is generally forested (natural) and reflected in the validation basins.

Please note that all hydrographs and performance estimates of the four experiments regarding streamflow performance can be found on the interactive website associated with this work (www.hydrohub.org/grip-gl/maps_streamflow.html). The users



can compare hydrographs at all locations for models of interest and zoom in to specific time periods/events. Follow up studies could look at the performance of subsets of models for specific events or under specific, e.g., low or high flow, conditions.

## 3.2 Basin-wise model performance regarding auxiliary variables

In this section the performance of models regarding additional variables at the basin-scale (comparable to streamflow) will be

analysed for twelve models. The thirteenth model, i.e. Machine-Learning based LSTM-lumped, is not (yet) able to provide estimates for those variables and will not be considered for the remaining analyses. The basin-wise comparison of additional variables implies that the reference datasets as well as the subbasin based and gridded model outputs were aggregated to basin-level. Once the outputs and reference datasets are aggregated in space, the resulting daily time series are compared. The 212 calibration and validation locations are compared without distinguishing them (as done for streamflow) since the

additional variables were never calibrated and their performances are therefore of "validation" quality. The results for the three additional variables are presented in Fig. 4. Each panel of that figure represents the results for one of the three variables and will be discussed one-by-one in the following three subsections (Sect. 3.2.1 to Sect. 3.2.3). The last subsection (Sect. 3.2.4) will summarize these results and discussions.

### 3.2.1 Basin-wise comparison of actual evapotranspiration

Regarding actual evapotranspiration (AET), the locally calibrated models exhibit on average a lower performance than the regionally calibrated, physically-based models (Fig. 4A).

The only exception is the HYMOD2-lumped model. During its development, a focus was on improving the realism of simulated AET which was achieved by introducing a model structure in HYMOD2-lumped that produced simulated AET results closely approximating GLEAM AET (Roy et al., 2017). In other words, the HYMOD2-lumped model was implicitly

pre-informed by this dataset which would explain the superior performance seen here. Note that all evaporation related (and other) parameters were re-calibrated using only streamflow observations in this study and were not reused from the development stage presented by Roy et al. (2017).

Besides the HYMOD2-lumped with a median KGE of 0.72, only regionally calibrated models show good performance levels (above KGE of 0.65), i.e. SWAT-Raven with a median KGE of 0.74, MESH-SVS-Raven with a median KGE of 0.68, and GEM-

Hydro-Watroute with a median KGE of 0.70. This is notable as these latter models showed consistently lower performances regarding streamflow compared to the locally calibrated models. Regarding MESH-SVS-Raven and GEM-Hydro-Watroute, it is possible that their performance regarding AET could be even slightly better once the bug present in MESH-SVS-Raven and affecting vegetation cover during this work is fixed (see Sect. 3.1.1). Note that this bug probably affected these two models to a lower degree for the other auxiliary variables.

The other models exhibit medium performance levels above KGE values of 0.48. Two exceptions of models with poor performance levels are GR4J-lumped with a median KGE of 0.44 and LBRM-CC-lumped with a median KGE of 0.15.

The significantly low performance of LBRM-CC-lumped was surprising. Subsequent to the posting of model results, an error was found in the calibration setup that resulted in ignoring an important constraint on the proportionality constant controlling

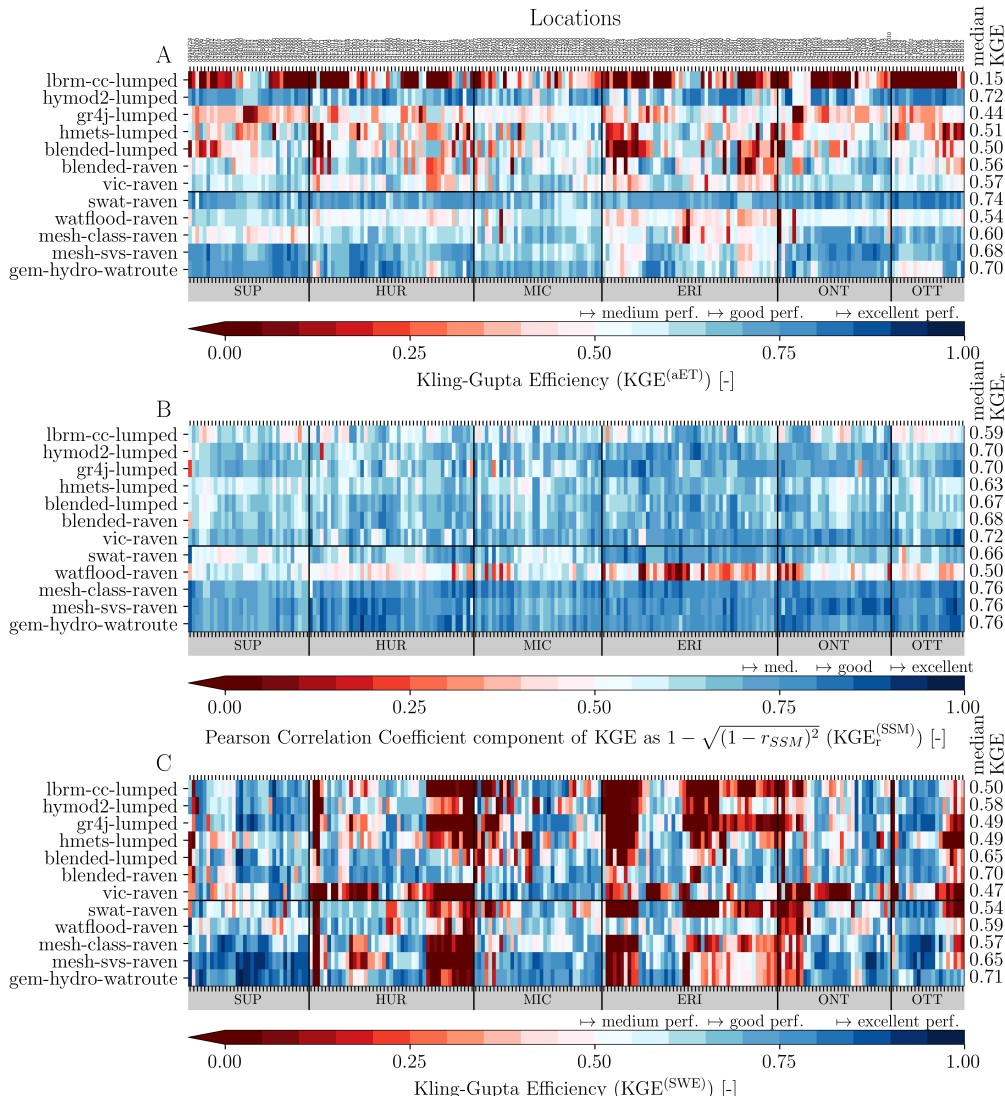

**Figure 4. Basin-wise model performance regarding auxiliary variables.** The performance of twelve models (y-axis) is shown for (A) actual evapotranspiration, (B) surface soil moisture, and (C) snow water equivalent. The model outputs are aggregated to the 212 basins (x-axis) used in this study (141 calibration locations and 71 validation locations). The performance is evaluated for observations available between 2001 and 2017 (overall period used for streamflow calibration and validation). The Kling-Gupta efficiency (KGE) is used to evaluate the simulations of actual evapotranspiration and snow water equivalent while the Pearson correlation is used for the (standardized) surface soil moisture. The colorbar in each panel is organized such that blue (red) colors indicate good (poor) performance. The machine-learning-based LSTM model is not simulating these additional variables and is hence not included. The basins (x-axis) are grouped into six regions of the Lake Erie (ERI), Lake Huron (HUR), Lake Michigan (MIC), Lake Ontario (ONT), Ottawa River (OTT), and Lake Superior (SUP) watershed. The basin order is the same in the three panels. The horizontal black line separates models that are calibrated locally from models that are calibrated per region. For the spatial distribution of these results as well as the simulated and reference time series please refer to the website (e.g., the website for actual evapotranspiration www.hydrohub.org/grip-gl/maps_aet.html).





PET (see Lofgren and Rouhana, 2016, for an explanation of this constraint). In addition, the LBRM-CC-lumped team has
found that an improved representation of the long term average temperatures applied in the PET formulation would improve
results in some watersheds (PET is a function of the difference between daily temperature and long term average temperature
for the day of year). Accordingly, this may improve the LBRM-CC-lumped performance in simulating AET in future studies.

Note that there are no clusters of stations/basins where several models perform exceptionally good or bad which in turn
means that differences between models are indeed model-specific and not caused by any general problem with data at certain
stations. Furthermore, there are no difference between regions, i.e., models in one region performing significantly better or
worse than in other regions, which means that models perform similar in all geographic regions no matter the amount of
human impact or landuse, or if stations where originally calibrated or only validated regarding streamflow.

### 3.2.2 Basin-wise comparison of surface soil moisture

The model performances regarding surface soil moisture are consistent both across models and locations (Fig. 4B). The per-
formance levels can be classified as poor ($KGE_r$ of lower than 0.7) for LBRM-CC-lumped (0.59), HMETS-lumped (0.63),
Blended-lumped (0.67), Blended-Raven (0.68), SWAT-Raven (0.66), and WATFLOOD-Raven (0.50). Even though classified
as "poor", correlation coefficients of above 0.6 for some largely conceptual models is surprising – especially given that most
not even explicitly simulate soil moisture internally but convert simulated water contents into proxy soil moisture estimates.

The other models show medium performance levels ($KGE_r$ of at least 0.7), i.e. HYMOD2-lumped (0.70), GR4J-lumped
(0.70), VIC-Raven (0.72), MESH-CLASS-Raven (0.76), MESH-SVS-Raven (0.76) and GEM-Hydro-Watroute (0.76). It is
notable that the four models that perform best are all physically-based models that explicitly simulate soil moisture as a state
variable.

WATFLOOD-Raven seems to be an outlier with its poor median $KGE_r$ performance of 0.50. Especially stations located in
the Lake Erie watershed show low correlation values compared to the reference dataset which can be explained by the fact that
WATFLOOD does not explicitly simulate SSM and therefore the rescaled Upper Zone Storage (UZS) was used as a proxy for
SSM. Specific parameter settings were chosen to mimic the behavior in urban areas (i.e., low value of infiltration coefficient
and low value of upper zone retention limiting UZS) and restrict fluctuations in the UZS as this would be expected for SSM
but these parameter setting are likely not enough to fully capture SSM dynamics in urban areas.

There are – similar to AET – no significant performance differences between regions. No basin can be identified as outlier
with significant lower or higher performance which indicates that all models are in good agreement with the GLEAM-v3.5 soil
moisture reference dataset and no inconsistency can be diagnosed.

### 3.2.3 Basin-wise comparison of snow water equivalent

The most diverse model performance across models and locations can be observed for snow water equivalent (Fig. 4C). One
model performs poorly (median KGE below 0.48), seven models show an overall medium performance (KGE at least 0.48),
while the other four models perform good (KGE at least 0.65).



The overall best performing models are GEM-Hydro-Watroute (median KGE of 0.71), Blended-Raven (median KGE of 0.70), MESH-SVS-Raven (median KGE of 0.65), and Blended-lumped (median KGE of 0.65). This represents an interesting mix of models as they are (a) physically-based and distributed (GEM-Hydro-Watroute, MESH-SVS-Raven), (b) conceptual and lumped (Blended-lumped) and (c) conceptual and distributed (Blended-Raven). It hence shows that even simplified, lumped,
and conceptual models have the potential to simulate basin-scale SWE with a similar quality as more complex, distributed models.

The models with the weakest performance are VIC-Raven (median KGE of 0.47), GR4J-lumped (median KGE of 0.49), HMETS-lumped (median KGE of 0.49), and LBRM-CC-lumped (median KGE of 0.50). Again, these are both conceptual/lumped and physically-based/distributed models. The weak performing models are exclusively locally calibrated and might
have suffered from poor donor-basin mappings. However, other locally calibrated models like the two Blended models used the same (potentially inappropriate) donor basins and show superior performance.

SWE is the only variable that reveals clear differences in performance between regions: The Lake Superior (SUP) watershed is the region where all models show good performance at all location whereas in other regions, e.g., in the Lake Huron (HUR) watershed, several locations show poor performance across most models. The locations with poor performance (red color
across most models) are located in the same geographical area, i.e., the land area south of Lake Huron, north of Lake Erie and West of Lake Ontario which is – besides being the most densely populated area of the Great Lakes watershed – heavily impacted by lake effects in precipitation and complex precipitation phase (Mekis et al., 2020) affecting snow accumulation and melt. Another reason for poor performances of the models in this region might be the quality of forcing datasets which are known to have problems in lake effect areas. It is not surprising that most models exhibit lower performances in these regions
compared to elsewhere.

It is however notable that some models, e.g., the locally calibrated Blended-Raven model and the regionally calibrated WATFLOOD-Raven model perform exceptionally well at locations where other models deteriorate in performance even though it does not use special snow processes but the basic formulation of the temperature index model (Anderson, 1973). The Blended-Raven model is not using any special algorithm to simulate snow processes. It indeed uses the same algorithmic
representation as the Blended-lumped model which is slightly worse at these locations. But the subbasin discretization of the Blended-Raven model (only difference to the Blended-lumped model) seems to improve performances at several locations.

### 3.2.4  Summary basin-wise comparisons of auxiliary variables

In summary, the common assumption that distributed, physically-based models are intrinsically of superior quality when evaluating model states beyond streamflow can not be confirmed in all cases.
For actual evapotranspiration, a locally calibrated, conceptual model (HYMOD2-lumped) shows exceptional performance on par with more complex models (i.e., SWAT-Raven, MESH-SVS-Raven and GEM-Hydro-Watroute). Even though for surface soil moisture physically-based models are generally best (i.e., MESH-CLASS-Raven, MESH-SVS-Raven, and GEM-Hydro-Watroute), the less complex and locally calibrated models (i.e., HYMOD2-lumped, GR4J-lumped, and VIC-Raven) can lead to comparable performances when SSM is standardized as described in Sect. 2.7 and evaluated based on its correlation with





observations. Indeed, when looking at the true KGE for actual SSM values, only MESH-SVS-Raven and GEM-Hydro-Watroute showed reasonable performances (not shown here), partly because SVS represents the actual SSM values over thin superficial soil layers. However, a physically-based model (WATFLOOD-Raven) shows an outlier poor performance across all locations for SSM. For snow water equivalent, there are as many good performing locally calibrated/conceptual models as there are physically-based models, i.e., Blended-lumped and Blended-Raven vs. MESH-SVS-Raven and GEM-Hydro-Watroute.

The analysis of basin-aggregate estimates might however favour models that are setup and run on this (basin) scale. The distributed/gridded models might be indeed superior when the sub-grid variability is analysed. Hence, the next section will focus on analysing the models on the native, gridded resolution of the reference dataset rather than aggregating models and reference dataset to basin-scale.

Note that all time series and metrics are available on websites associated with this study. The websites are especially insight-
ful as they provide a spatial representation of the stations which is not available in Fig. 4. On the websites the users will be able to identify the region where, for example, the weak performing snow water equivalent basins are located. There is one website for each additional variable analysed here, i.e., actual evapotranspiration (www.hydrohub.org/grip-gl/maps_aet.html), surface soil moisture (www.hydrohub.org/grip-gl/maps_ssm.html), and snow water equivalent (www.hydrohub.org/grip-gl/maps_swe. html).

### 3.3 Grid-cell-wise model performance regarding auxiliary variables

In this section the results of the model comparisons for the three additional variables are presented. In contrast to the previous section, the variables are compared at the grid-cell scale rather than per basin. This means, all model outputs were regridded to the grid of the corresponding reference dataset. The performance metrics are then compared at each grid cell leading to maps of performances presented in Fig. 5 for actual evapotranspiration, Fig. 6 for surface soil moisture, and Fig. 7 for snow water
equivalent. The results will be presented for only twelve models as the Machine Learning-based LSTM-lumped model did not simulate these additional variables. The results will be presented and discussed in the following three subsections (Sect. 3.3.1 to Sect. 3.3.3) while the last subsection summarizes the findings (Sect. 3.3.4).

#### 3.3.1 Grid-cell-wise comparison of actual evapotranspiration

The results of the grid-cell-wise comparisons regarding actual evapotranspiration (Fig. 5) are consistent with the basin-wise
comparisons discussed before (see Sect. 3.2.1 and Fig. 4A). The best models (visually and by numbers) are the locally calibrated HYMOD2-lumped and the regionally calibrated SWAT-Raven. Other models with good performance are MESH-SVS-Raven and GEM-Hydro-Watroute while the weakest models are the two locally calibrated LBRM-CC-lumped and GR4J-lumped models. Note that the median performance per model is not exactly the same as the performance value for the basin-wise comparison (Sect. 3.2; Fig. 4). This is due to the fact that the basin-wise performances are derived as the median over 212
basins while in the grid-cell-wise comparisons they are derived over the 1213 grid-cells that appear in the reference dataset as well as being available in all 12 model outputs.





The grid-cell-wise analysis provides more insights into where results are good or weak spatially. In general, the locally calibrated models (LBRM-CC-lumped to VIC-Raven) expectedly exhibit discontinuous (patchwork) patterns of performance while the regionally calibrated models (SWAT-Raven to GEM-Hydro-Watroute) are more seamless – at least within each of
the six main regions. The discontinuous nature of model outputs (and hence their performance when compared to continuous fields of observations) has been reported in literature before (e.g., Mizukami et al., 2017; Yang et al., 2019).

The locally calibrated LBRM-CC-lumped model shows poor performance (median KGE of 0.14) in large regions (likely to improve once the error in the calibration setup is resolved and a better representation of long term average temperatures is in place; see details in Sect. 2.4.2). The HYMOD2-lumped model performs (very) good (median KGE of 0.76) everywhere
(almost no red cells in entire domain). The GR4J-lumped model exhibits a poor performance (median KGE of 0.39) with deficiencies especially in Northern latitudes but also south of Lake Ontario and Lake Erie. The HMETS-lumped model with its medium performance level (median KGE of 0.53) with patchwork performance due to its local calibration and transfer of parameter sets to validation basins. The latter are the reason for the patchy red regions especially in the Ottawa River basin as that region might not have had enough basins for training (as discussed earlier). The Blended-lumped and Blended-Raven
models exhibit both medium performance levels (median KGEs of 0.58 and 0.56, respectively) with their spatial patterns of performance being similar. This is expected given the very similar nature of these two models. The patterns are patchy but overall good except in validation basins which might be caused by a (potentially) inappropriate donor basin being assigned. The VIC-Raven model's medium performance (median KGE of 0.57) is lowest in validation basins and west and north of Lake Erie. The latter might be due to the models weakness of simulating AET over urban areas.

The regionally calibrated SWAT-Raven exhibits a median KGE of 0.73 and hence performs comparably good as the locally calibrated HYMOD2-lumped with almost no weak (red) performing grid cells. The WATFLOOD-Raven shows a medium performance level (median KGE of 0.56) with weakest performance around Lake Erie (highly urban and agricultural) and within the Ottawa River watershed (potentially caused by not enough basins for training). The medium performing MESH-CLASS-Raven model (median KGE of 0.60) shows similar behaviour in Lake Erie watershed as the WATFLOOD-Raven
and the MESH-SVS-Raven model. It has further some deficiencies around Lake Superior. The MESH-SVS-Raven model is performing good (median KGE of 0.66) with lowest performance in agricultural heavy regions south-west of Lake Erie. The GEM-Hydro-Watroute with its good overall performance (median KGE of 0.66) shows the weakest performances over (a) the Ottawa River watershed due to limited training data/basins in this region which were not sufficient to calibrate the model in this region appropriately and (b) the agricultural regions south-west of Lake Erie. Again, MESH-SVS-Raven and GEM-
Hydro-Watroute could probably have a slightly better AET performance if the bug present in MESH-SVS-Raven and affecting vegetation cover during this work was fixed (see Sect. 3.1.1).

### 3.3.2   Grid-cell-wise comparison of surface soil moisture

The results of the comparison of grid-cell-wise model outputs with the gridded reference dataset of surface soil moisture (Fig. 6) are consistent with the results previously discussed for the basin-wise comparisons (see Sect. 3.2.2 and Fig. 4B) but allow for
more detailed analysis of regions of poor performance which might help to point to potentials for model improvements.

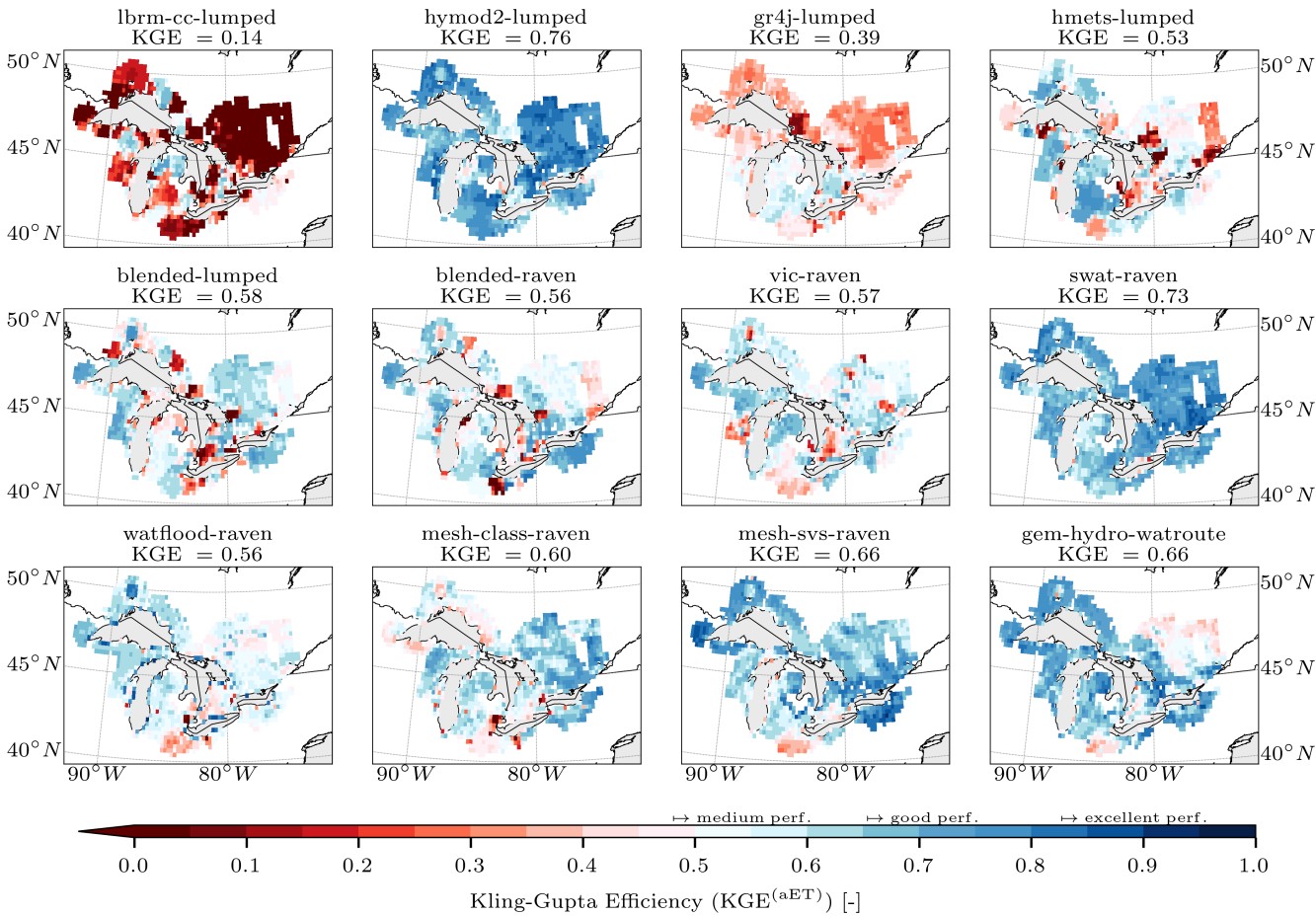

**Figure 5. Grid-cell-based model performance regarding actual evapotranspiration.** The performance of twelve models (panels) regarding actual evapotranspiration (aET) is shown. The model outputs are regridded to the grid of the reference dataset (here GLEAM v3.5b; variable E). Only grid cells that are available in the reference dataset and all twelve models are displayed (1213 grid-cells in total) and used to derive the Kling-Gupta Efficiency (KGE) metrics based on all available observations between January 2001 and December 2017. The median KGE across space is added as a label on top of each panel. The colorbar has labels added to categorize the performance into medium, good, and excellent. The machine-learning-based LSTM model is not displayed as this is the only model not simulating actual evapotranspiration explicitly.



Note that the median performances per model are not the exact same values as the ones reported for the basin-wise comparison since those are derived as the median over 212 basins while here they are derived over the 1213 grid-cells that appear in the reference dataset and are available in all 12 model outputs. Also note that the following discussion of results will list the individual models in reverse order compared to their appearance in the panels of Fig. 6 due to the more natural flow of the
discussion.

The regionally calibrated MESH-CLASS-Raven, MESH-SVS-Raven, and GEM-Hydro-Watroute models are visually (and by numbers) best performing. Their performance levels (median $KGE_r$ of 0.73 , 0.71, and 0.72, respectively) can be classified as medium. The three spatial performance patterns look very similar which can be explained by the fact that CLASS and SVS share similarities regarding soil moisture processes. The soil moisture estimates are especially poor around Lake Erie and
within the parts of the Ottawa River watershed that were not calibrated (North-Eastern part of the Great Lakes basin), because of the persistence of the variability in soil properties in this region. The other regionally calibrated model SWAT-Raven is performing well in Southern parts of the Great Lakes watershed and has a lower performance in Northern parts of the Ottawa River watershed and the Lake Superior region.

Most locally calibrated models show similar performances (median $KGE_r$ of 0.62 to 0.65) with some patchwork pattern
of weak performances especially in regions that are potentially located in validation basins. The HMETS-lumped has an overall consistently lower performance (median $KGE_r$ of 0.59) compared to the Blended-lumped and Blended-Raven models which is interesting as the blended model structure of the those two models was built upon the HMETS-lumped model. The improvement shows the added value of the blended structure upon the original fixed HMETS-lumped model structure. The LBRM-CC-lumped model has the overall weakest performance mostly due to its lower performance across the Ottawa River
basin which is consistent with the weak performance of this model regarding actual evapotranspiration discussed earlier. Several avenues have been identified after posting the model results for LBRM-CC-lumped that may improve the LBRM-CC-lumped performance in future studies.

### 3.3.3 Grid-cell-wise comparison of snow water equivalent

The performance of the grid-cell-wise model outputs compared to a gridded reference dataset of snow water equivalent (Fig. 7)
are overall consistent with the results previously discussed for the basin-wise comparisons (see Sect. 3.2.3 and Fig. 4C). The results, however, highlight the large variability of performances in space and across models.

Note that the median performances per model are not the exact same values as the ones for the basin-wise comparisons. The latter are derived as the median over 212 basins while here they are derived over the 6174 grid-cells that appear in the reference dataset and are available in all 12 model outputs.
In contrast to the two previous variables analysed, the difference between the basin-wise and the grid-cells-wise comparisons are significant for some models. We label "significant" as differences equal or larger than 0.05 in median KGE. All significant differences are improvements compared to the basin-wise performances, i.e. LBRM-CC-lumped increases performance from a median KGE of 0.50 to 0.59, GR4J-lumped from 0.49 to 0.65, VIC-Raven from 0.47 to 0.58, SWAT-Raven from 0.54 to 0.61, MESH-CLASS-Raven from 0.57 to 0.71, and MESH-SVS-Raven from 0.65 to 0.74. This behavior can be explained as follows:



**Figure 6. Grid-cell-based model performance regarding surface soil moisture.** The performance of twelve models (panels) regarding surface soil moisture (SSM) is shown. The model outputs are regridded to the grid of the reference dataset (here GLEAM v3.5b; variable `SMsurf`). Only grid cells that are available in the reference dataset and all twelve models are displayed (1213 grid-cells in total) and used to derive the Pearson correlation coefficients ($KGE_r$) based on all available summer month observations (June to October) between 2001 and 2017. The median $KGE_r$ across space is added as a label on top of each panel. The colorbar has labels added to categorize the performance into medium, good, and excellent. The machine-learning-based LSTM model is not displayed as this is the only model not simulating surface soil moisture explicitly. Note that only VIC-Raven, SWAT-Raven, MESH-CLASS-Raven, MESH-SVS-Raven, and GEM-Hydro simulate (surface) soil moisture explicitly while the remaining models used proxy variables for soil moisture. All SSM estimates were standardized to account for the proxy nature of the variables (Eq. 5) and, hence, only the correlation coefficients ($KGE_r$) is reported here.





In total, 193 of the 212 basins used in this study have similar drainage areas between 300 km$^2$ and around 5000 km$^2$, 11 basins have a drainage area between 5000 and 10,000 km$^2$, 6 basins are between 10,000 and 20,000 km$^2$ in size and the two largest basins have a drainage area of 25,000 and 90,000 km$^2$. The second-largest basin is covering the entire most Northern part of the Lake Superior watershed and is draining towards streamflow gauge 02AD012 while the largest basin is draining towards streamflow gauge 02KF005 within the Ottawa River basin. It is covering large parts of the domain east of the Georgian Bay and

North of Lake Ontario (see website for drainage domain of all streamflow locations). The largest (90,000 km$^2$) basin covers around 15% of the entire domain analysed in the grid-cell-wise comparisons; the second largest basin (25,000 km$^2$) covers around 4% of the domain. Hence, if a model is performing well in theses areas, it will contribute to 19% of the entire KGE "score" while in a basin-wise comparison it is only two of 212 basins which equals to around 1% of the overall score. It can be seen in Fig. 7 that models that have good (blue and dark blue) performances in, for example, the Ottawa River watershed

(North-eastern part of the Great Lakes watershed), exhibit a significant gain regarding the median KGE performance when derived per grid-cell.

   This result emphasizes the importance of decisions made to evaluate model performances. Basin-wise comparisons seem to be easier to communicate and derive – especially for a wide range of models with different underlying model discretizations. But comparisons that use the native resolution of reference datasets (here gridded) might be more appropriate. We showcase

here that even models that are not setup in gridded mode, i.e., lumped and subbasin based models can be converted to a grid-scale to match the resolution of the reference datasets. The results also show that lumped models not necessarily deteriorate when compared on a grid-cell scale. The results indeed suggest quite the contrary since, for example, models like LBRM-CC-lumped and GR4J-lumped increase their median KGE score because they predicated the snow water equivalent accurately in these large basins.

The detailed analysis of the models (again in reverse order) shows that the regionally calibrated GEM-Hydro-Watroute, MESH-SVS-Raven and MESH-CLASS-Raven models perform equally and exhibit a good performance level (median KGE of 0.72, 0.74, and 0.71, respectively). Main areas of weak performance are the most densely populated area of the study domain, i.e., the region north and west of Lake Erie and South of Lake Huron/Georgian Bay. The WATFLOOD-Raven shows a medium, patchy performance overall (median KGE of 0.58). No specific areas of weak or excellent performance can be

identified. SWAT-Raven's medium overall performance (median KGE of 0.61) pattern looks similar to the two MESH and the GEM-Hydro-Watroute models with overall larger and more pronounces weaker areas.

   The locally calibrated VIC-Raven with its overall medium performance (median KGE of 0.58) shows a different pattern of weak regions. The low-performing regions are now mostly located in the Ottawa River basin and east of Lake Superior. The weak performing regions seem to be mostly forested which might indicate that the calibration based on only streamflow is

not properly constraining parameters governing the evolution of snow in forested environments in VIC. The Blended-lumped and Blended-Raven models show similar good performance compared to each other (median KGE of 0.66 and 0.68, respectively) while the distributed Blended-Raven is slightly superior in most regions (especially west of Lake Michigan and west and north of Lake Erie). This highlights again the added value of a distributed model setup compared to its lumped counterpart (at least for the Blended model). The HMETS-lumped model with its medium overall performance (median KGE of 0.53)





shows similar weak regions but more pronounced (weaker) as the two Blended models which, again, demonstrates the added
value of the blended models structure as the Blended models are based upon the HMETS model with adding structural flexibil-
ity. The LBRM-CC-lumped, HYMOD2-lumped and GR4J-lumped models show comparable medium performances (median
KGE of 0.59, 0.62, and 0.65, respectively) with weakest performances in the Southern part of the Great Lakes watershed and
consistently better performances above around 46°N.

Most notable is that even though the most complex, physically-based models (MESH-CLASS-Raven, MESH-SVS-Raven,
and GEM-Hydro-Watroute) show the overall best performance, some locally calibrated and more conceptual models are not
significantly weaker (e.g., Blended-Raven). Especially in regions where the physically-based models are weak (e.g., corridor
between Lake Huron and Lake Erie) those conceptual, local models perform much better. There is a potential that, when
more sophisticated approaches are applied to assign donor-basin mappings, these locally calibrated models might significantly
improve relative to the more complex models.

### 3.3.4 Summary grid-cell-wise comparisons of auxiliary variables

The grid-cell-wise comparisons lead in general to similar conclusions as the basin-wise evaluation with more detailed insights
regarding spatial performances distributions. The main conclusion – as seen for the snow water equivalent variable – is that one
needs to be aware of the the underlying assumptions when results are compared on a basin-scale, i.e. when outputs are spatially
aggregated before the comparison. This leads to situations where (a) large basins will have the same "weight" when compared
to smaller basins and (b) large variability in performances within a basin can cancel each other out when aggregated in space
before the performance is evaluated. The impact of this depends on the resolution and spatial variability of the reference dataset;
it seems that actual evapotranspiration and surface soil moisture have more robust, i.e. similar on basin- and grid-scale, while
snow water equivalent led to large differences between the two methods of comparison.

In contrast to basin-wise comparisons, the grid-cell-wise comparisons enhance analysis of spatial inconsistencies (patchwork
patterns) which are most apparent in locally calibrated models but to some extent also visible for regionally calibrated models.
The spatial analysis improves the identification of reasons of weak model performances due to the more intuitive comparison
with, for example, landuse types or urban areas. This identification of regions with poor performance might lead to more
targeted model improvements– especially when other models are able to simulate variables in those regions. The regions of
weak performance for the auxiliary variables and models used here can be explained with poor donor basin selection, urban
and agricultural landuse, and regions for which not enough calibration basins were used during training (Ottawa River region).

It can not be confirmed that physically-based models are always better performing than more conceptual models. It might
be beneficial to use locally calibrated, simplified models to augment physically-based models.

The variability of model performance (meaning the degree of patchwork patterns) looks reasonable even for lumped models;
no large discontinuities can be observed. This is most likely be caused by the large set and good spatial distribution of locations
(basins) used for training. A smaller set or more clustered selection of training basins would have led to larger regions of weak
performance; as can be seen of the Ottawa River basin where this indeed impacted the overall performance. We encourage
follow-up studies looking into more local comparisons of the model outputs derived here and provided as open datasets.

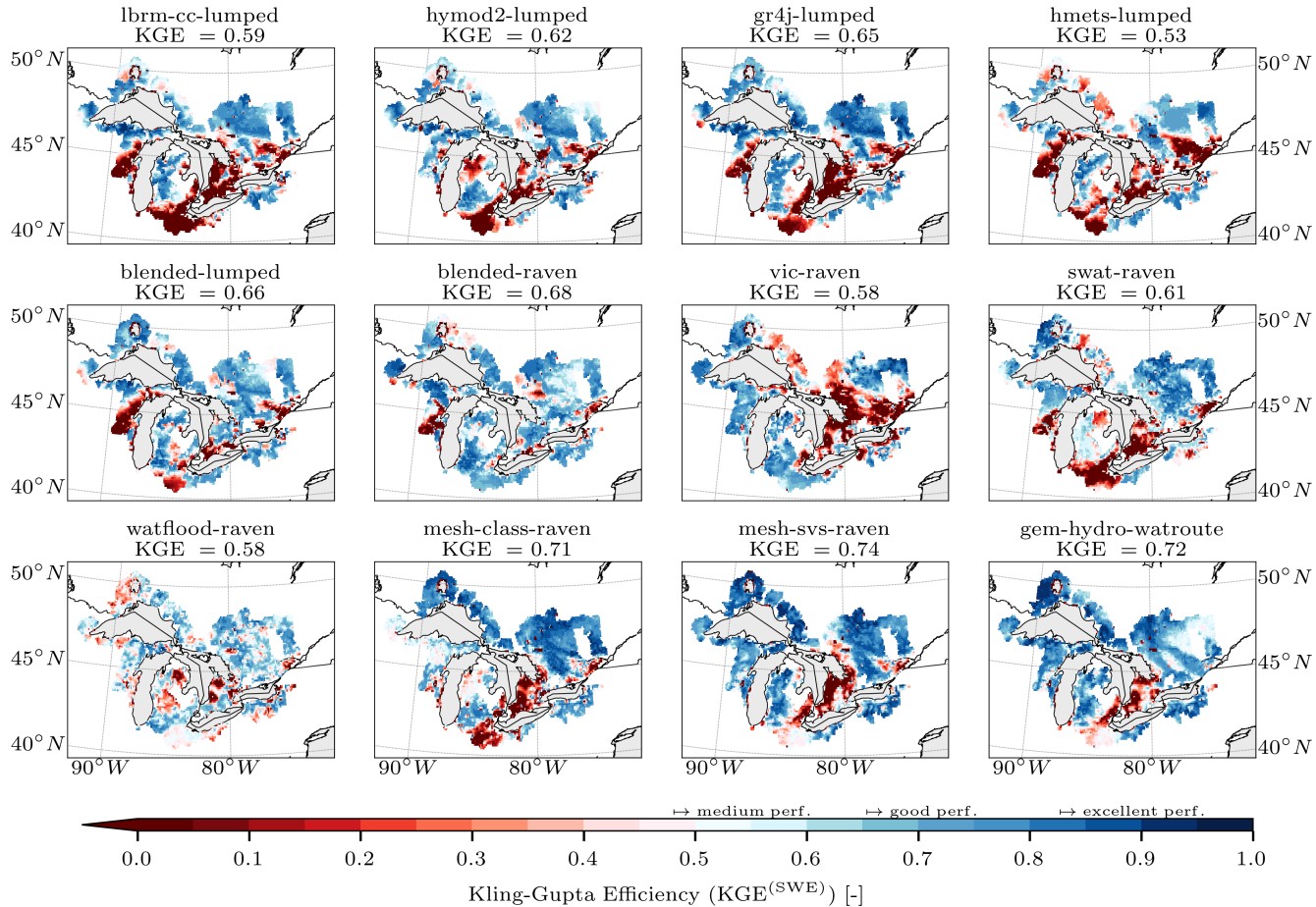

**Figure 7. Grid-cell-based model performance regarding snow water equivalent.** The performance of twelve models (panels) regarding snow water equivalent (SWE) is shown. The model outputs are regridded to the grid of the reference dataset (here ERA5-Land; variable `sd`). Only grid cells that are available in the reference dataset and all twelve models are displayed (6174 grid-cells in total) and used to derive the Kling-Gupta Efficiency (KGE) values based on all available observations larger than 1 mm of daily SWE between January 2001 and December 2017. The median KGE across space is added as a label on top of each panel. The colorbar has labels added to categorize the performance into medium, good, and excellent. The machine-learning-based LSTM model is not displayed as this is the only model not simulating snow water equivalent explicitly.





### 3.4 Multi-objective multi-variable analysis of model performances

The multi-objective analysis of model performances is to consolidate the results of the models that provided simulated stream-flow and the three additional variables (Fig. 8). Hence, the Machine Learning-based LSTM-lumped model is not included here as it only estimated streamflow. The performances at the basin-scale are evaluated to make sure all four variables are handled equally (no grid-cell-wise performance metrics). In the same vein, only streamflow simulations of the validation period are used ensuring the Pareto analysis is based only on validation/evaluation data.

The results are shown for using all four variables (Fig. 8A) as well as for groups of three variables each where one of the additional variables AET, SSM, and SWE is discarded (Fig. 8B, Fig. 8C, and Fig. 8D, respectively). The latter three are included to discuss whether a good (or weak) model performance is exclusively due to a superior performance of one of those additional variables.

The results show that the locally calibrated models perform better (without any exception) for the calibration basins (orange 1070 bars) compared to the validation basins (red bars). Exactly the opposite is happening for the regionally calibrated models (light blue bars for calibration basins vs dark blue bars for validation basins), which highlights the stronger spatial robustness of regionally or globally calibrated frameworks, even though the parameter transfer strategy employed here with locally calibrated models could be improved. Only when surface soil moisture is not included, the performance of two regionally calibrated models (MESH-SVS-Raven and GEM-Hydro-Watroute) drops for the set of validation basins.

The results including all 4 variables (Fig. 8A) indicate that HYMOD2-lumped is the locally calibrated model (red colored bars) with the highest overall performance followed by the Blended-Raven and the VIC-Raven model. The best overall regionally calibrated model is GEM-Hydro-Watroute followed by MESH-SVS-Raven. MESH-CLASS-Raven is third-best in calibration but is outperformed by SWAT-Raven for the validation basins due to SWAT-Raven's significant improvement for validation basins. Models of overall poor performance are the two locally calibrated LBRM-CC-lumped and HMETS-lumped 1080 as well as the regionally calibrated WATFLOOD-Raven.

In this study, HYMOD2-lumped and SWAT-Raven achieve their maximum performance when their exceptionally good performance for actual evapotranspiration (AET) is taken into account, as can be seen by their significant performance loss (both -39%) when this variable is not included in the multi-objective analysis (Fig. 8B). This means that these two models are usually part of the Pareto front of non-dominated models because no other model is as good in simulating AET. Similarly, 1085 the performance of the following model would have been significantly lower if surface soil moisture (SSM) would not have been included in the analysis (Fig. 8C): GR4J-lumped (-25%), MESH-CLASS-Raven (-36%), MESH-SVS-Raven (-29%), and GEM-Hydro-Watroute (-26%). This in turn means that these models perform comparably better than other models regarding surface soil moisture. A significant drop of performance can be detected for the Blended-Raven model (-25%) when snow water equivalent is not considered (Fig. 8D) showing that this model outperforms several others for snow estimates.

The models which overall performance is mainly due to a single specific variables' performance (HYMOD2-lumped and SWAT-Raven for AET, MESH-CLASS-Raven or MESH-SVS-Raven or GEM-Hydro-Watroute for SSM, and Blended-Raven for SWE) are considered good starting points to study in order to improve models with lower performance for these variables.



This is due to the fact that the removal of a variable and a simultaneous drop in performance of a model, can only be caused by this model often dominating other models regarding this variable. Note that the models with good performance for specific
variables listed above are not necessarily the models that are the best for this variable (Sec. 3.2), e.g., GEM-Hydro-Watroute is slightly better than Blended-Raven for SWE (see Fig. 4C; median KGE values of 0.71 and 0.70, respectively). However, there is a significant amount of basins where Blended-Raven is better than the GEM-Hydro-Watroute model (and most other models) especially in the Lake Huron (HUR) and Lake Erie (ERI) watershed. This leads to the fact that the Blended-Raven is on the Pareto front for all those basins since it is outperforming so many other models in those basins even though GEM-
Hydro-Watroute is better (by median). Even though GEM-Hydro-Watroute can definitely be considered state-of-the-art for SWE simulations one might want to study why Blended-Raven is so much better in those basins. The Pareto analysis was performed to reveal results like this.

The four Pareto analyses collectively show that three models appear notably less frequently in the group of non-dominated models: LBRM-CC-lumped, HMETS-lumped, and WATFLOOD-Raven. They show deficiencies in more than one variable
(e.g., multiple instances where the frequency they are non-dominated falls between 10% and 35% of the 212 basins). Using these models as is for additional applications in the Great Lakes region without for revisions/recalibration would warrant careful justification.

Locally calibrated models – especially HYMOD2-lumped, Blended-Raven, and VIC-Raven – show equivalent (if not better) results for calibration basins compared to the best regionally calibrated models, but generally significantly lower performances
for validation basins. The performance of locally calibrated models is likely to improve for validation basins with a revised donor basin mapping.

## 4    Conclusions

We present a carefully designed framework to compare models of different types (lumped, gridded, and subbasin-based; conceptual and physically-based; locally, regionally, and globally calibrated) consistently. The framework includes using the same
data to setup and force the models, using (mostly) the same routing, and the same methods of model performance estimation regarding streamflow and additional variables such as actual evapotranspiration (AET), surface soil moisture (SSM), and snow water equivalent (SWE). The strong focus on standardizing datasets and methods leads to observed performance differences that are to a large extent specific to the model or specific to subjective model building decisions (e.g., calibration details) rather than performance differences between models caused by input data of different quality or inconsistencies in post-processing
simulation outputs.

The majority of modelling teams learned about a poor assumption they made, a data processing error or a bug in their model code as a direct result of them interacting with other study co-authors not on their modelling team. First of all, this is evidence of the value of model intercomparison studies for collectively improving hydrological model quality. Second of all, this suggests that whenever possible, model intercomparisons should be iterative so as to enable teams to make changes to

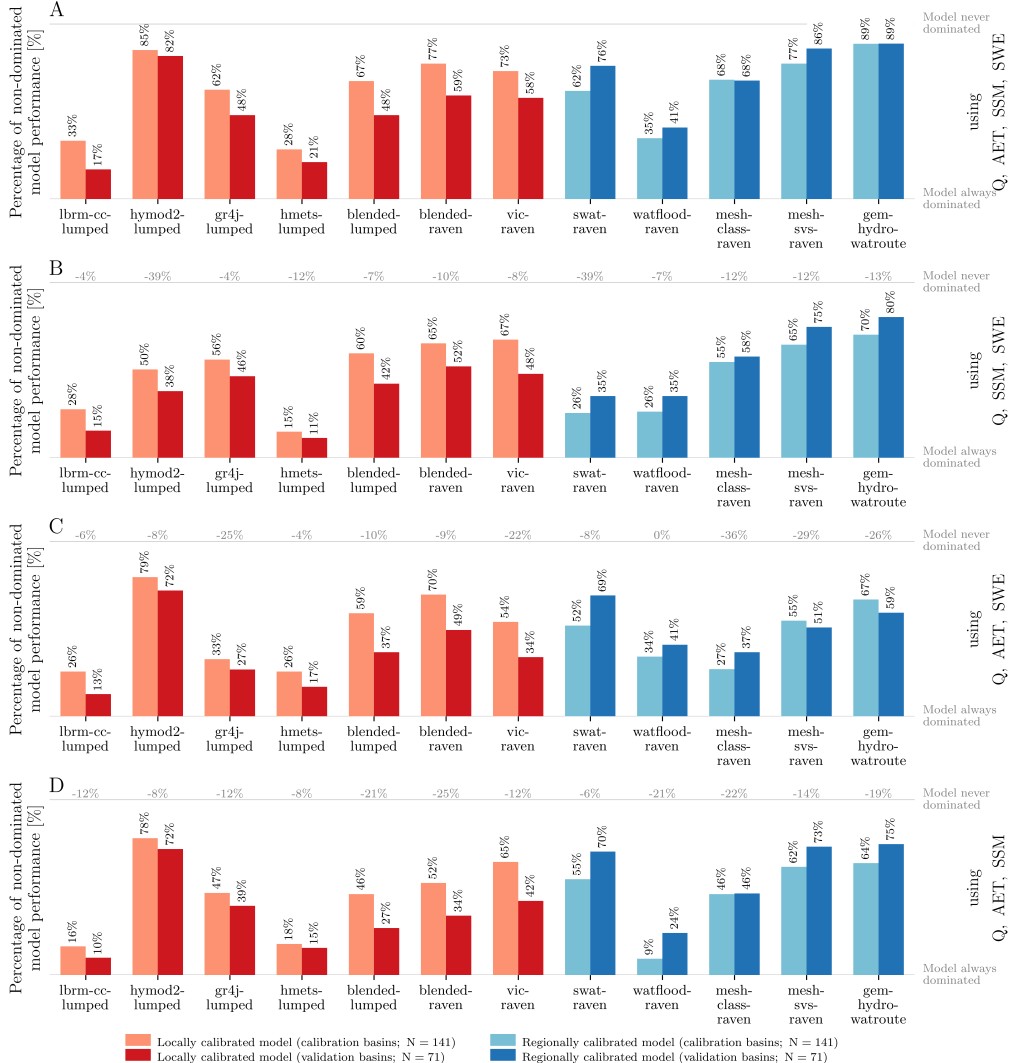

**Figure 8. Multi-objective analysis of model performances.** The twelve models (x-axis) that provided streamflow (Q) simulations and re-sults for the three additional variables – actual evapotranspiration (AET), surface soil moisture (SSM), and snow water equivalent (SWE) – are analysed using a multi-objective analysis to determine the number of times a model is superior in at least one of the considered performance metrics. This measure of non-dominance (y-axis) is derived using (A) all four variables (Q, AET, SSM, and SWE), (B) considering all vari-ables but not AET, (C) all variables but not standardized SSM, and (D) all variables except SWE. The metrics are derived for time periods not used during calibration, i.e. 2011 to 2017 for streamflow and 2001 to 2017 for all other variables. The estimates are derived using the basin-wise performances presented in Fig. 3 and Fig. 4. The derived percentages of non-dominance are added as labels above each bar in black. The percentages are derived for calibration basins (lighter colored bars) and validation locations (darker shaded bars). Panels (B-D) additionally show a gray labeled percentage value on top of the gray horizontal line which indicated perfect model skill. Those gray colored percentage values indicate how the model skill changed compared to the skill when all four variables are used (panel A). The models are grouped into models that locally calibrated each basins independently (red bars) and models that calibrated all locations of each region simultaneously (blue bars).





their model/approach. This is of course a challenge given most study co-authors received no funding explicitly supporting their participation in the model intercomparison.

Overall, this study generates a valuable model intercomparison dataset (fully documented input and model output datasets). This allows new models to be added to the intercomparison by future researchers. Just as important, this rich dataset is available for follow-up studies focused on learning why we saw the differences we did (e.g., given dominant LSTM-lumped performance for streamflow, which LSTM model build decisions were most important?). Our analysis and conclusions below are necessarily focused on reporting observed performance similarities and differences among the models.

The metrics for evaluating the model performance for each of the variables are presented as well as a multi-objective based Pareto method to integrate all performance metrics proposing an objective approach to evaluate model across all four variables leading to insights for specific models and groups of models. Furthermore, not only the calibration performance of the models regarding streamflow but also the performances in three different validation experiments (temporal, spatial, and spatio-temporal validation) are produced. The additional variables (AET, SSM, and SWE) are evaluated for all stations and the entire time period at once as the models were not trained using those data.

The results show that the globally calibrated Machine Learning based LSTM-lumped model is superior regarding streamflow in all experiments. The model does however not simulate any of the additional variables (AET, SSM, and SWE) as it is purely data driven and not calibrated using data other than streamflow. It is hence not possible to evaluate the overall model performance (based on all four variables). It is likely that already a small amount of additional data would suffice to fit the model to these additional variables (Lees et al., 2021b).

The locally calibrated (and mostly conceptual) models – with their advantage of being easy to setup and computationally efficient to allow for high-budget calibration experiments – show good performances in calibration and temporal validation and medium performances as soon as streamflow is evaluated at stations that have not been calibrated. Interestingly, it can be shown that the worst performing locally calibrated model is performing on average better than the best regionally calibrated (more complex and physically based) models regarding streamflow. Furthermore, the locally calibrated models are not necessarily performing worse than physically-based models when evaluated for the three additional variables (AET, SSM, and SWE), which, for SSM, is however due to the standardization of the variable and to the fact that it was evaluated based on correlation in this study, since most conceptual models do not explicitly represent actual SSM values for superficial soil. The top performing models, when considering all four variables analysed here, are GEM-Hydro-Watroute, HYMOD2-lumped and MESH-SVS-Raven. It is also clear that regionally calibrated models show a stronger spatial robustness than locally-calibrated models, when looking at the multi-objective evaluation of the four variables and comparing the drop of performances between calibration and validation basins. It is however expected that for locally calibrated models, the spatial validation performance and the performance regarding additional variables can be improved with a more sophisticated donor-basin-mapping strategy which is the objective of an ongoing follow-up study. Unless such improved transfer methods are in place, it seems advisable to not use locally calibrated models for ungauged basins as their spatial transfer can lead to poor performances at single gauges (in contrast to the median performance across many stations as analysed here).





The regionally calibrated (and physically-based) models are advantageous as they are setup over an entire region and simu-
lated parameter fields (including the additional variables) are more realistic (i.e., continuous) since they do not exhibit the patch-
work pattern caused by local setups. These setups also allow for a seamless prediction of streamflow at ungauged/untrained
locations and hence overcome the problem of the donor-basin mapping intrinsically. These models are however more time
consuming to setup and calibrate. The results show that regionally calibrated models are among the top performing models
(e.g, GEM-Hydro-Watroute and MESH-SVS-Raven) when all four variables are considered while they are consistently out-
performed by the locally calibrated models regarding streamflow, even if streamflow performances between the best regionally
calibrated and locally calibrated models become much closer when looking at validation basins over the validation period
(corresponding to the most difficult spatio-temporal validation evaluation performed here). Most regionally calibrated models
show deficiencies in basins that contain highly managed waterbodies, urban areas as well as in regions with predominantly
agricultural land use, especially on the US side of the Great Lakes. Limited training data is another reason for weaker model
performance as shown for the Ottawa River basin. It is expected that physically-based distributed models could therefore
strongly benefit from additional process representation (e.g. tile drains in agricultural areas, artificial drainage networks in ur-
ban areas, etc.) as well as from a better subdivision of a large domain to perform regional calibration, for example by calibrating
basins with a similar dominant land-cover type simultaneously (e.g. agricultural, urban, natural) rather than by calibrating very
different basins in the same region at the same time. More work is needed to assess the feasibility and efficiency of such a
methodology, however. Regarding water management, the representation of reservoir regulation, which is generally done in
the routing scheme directly but could probably be integrated inside Machine Learning based models as well, could improve
performances for heavily regulated watersheds.

Top performing models, when all variables are considered together, are the locally calibrated HYMOD-lumped (closely
followed by Blended-Raven and VIC-Raven) as well as the regionally calibrated GEM-Hydro-Watroute and MESH-SVS-
Raven (closely followed by SWAT-Raven and MESH-CLASS-Raven) while least performing models are LBRM-CC-lumped,
HMETS-lumped and WATFLOOD-Raven. From the multi-objective analysis including the additional variables (AET, SSM,
and SWE; Sect. 3.4), we learned that some models perform exceptionally well for additional variables compared to other
components of that model (HYMOD2-lumped and SWAT-Raven for AET; MESH-CLASS-Raven, MESH-SVS-Raven and
GEM-Hydro-Watroute for SSM; and Blended-Raven for SWE). When looking at the specific performances for each individ-
ual additional variable (Sect.s 3.2 and 3.3), the best models are: HYMOD2-lumped, SWAT-Raven, MESH-SVS-Raven, and
GEM-Hydro-Watroute for AET; HYMOD2-lumped (based on basin-wise only), GR4J-lumped (based on basin-wise only),
VIC-Raven (based on basin-wise only), MESH-CLASS-Raven, MESH-SVS-Raven, and GEM-Hydro-Watroute for SSM; and
Blended-lumped (based on basin-wise only), Blended-Raven, MESH-CLASS-Raven (based on grid-cell-wise only), MESH-
SVS-Raven, and GEM-Hydro-Watroute for SWE).

The comparison of the two approaches to determine the model performance regarding gridded additional variables, i.e.
grid-cell-wise and basin-wise comparison, shows that one needs to be aware of underlying assumptions when aggregating
gridded/distributed model outputs spatially. This especially holds when basins are of very different size or variables exhibit
large variability in space (such as SWE).



Across all models, it can be shown that most models show lower model performances (across all variables) in urban and
agricultural regions and in regions where snow is difficult to simulate due to urban areas and strong lake effects (e.g., eastern
shoreline regions of Lake Huron, Lake Erie and Lake Ontario).

The results and data produced in this study are made available through an interactive website to encourage follow up studies
and facilitate the communication of the results with a broad audience (beyond the hydrologic modeling community) as well as
encouraging the use of those data for more detailed analyses and follow up studies.

Future work will focus (among others) on testing more advanced donor basin mappings for the locally calibrated models,
compare the model outputs for additional variables at regions where performance is low in order to identify reasons behind the
poor performance, fixing identified bugs in several models and/or improving calibration methods leading to improved models,
adding additional models to the website once they become available, and calibration efforts including the additional variables
besides streamflow to further improve model realism.

*Code and data availability.*   The code and data used for this analysis will be made available on GitHub (https://github.com/julemai/GRIP-GL)
upon publication of the manuscript. Please note that the access of the GitHub repository will lead to a 404 error when accessed without per-
mission. The basin-scale results of this study are available under http://www.hydrohub.org/mips_introduction.html#grip-gl. Please note that
download of the data will not be activated before the final publication of this manuscript. BasinMaker used for producing the routing product is
available at http://hydrology.uwaterloo.ca/basinmaker/index.html. The Raven hydrologic modeling framework used for routing in most mod-
els and for the simulation of GR4J-lumped, HMETS-lumped, Blended-lumped, and Blended-Raven is available at http://raven.uwaterloo.ca.

*Author contributions.*   **All co-authors** provided feedback to the manuscript and participated on a regular basis in the monthly project meet-
ings where we discussed experimental setups and results. Beyond that the co-authors contributed the following: **Juliane Mai** managed the
project, designed experiments, selected streamflow gauging stations, prepared input data, processed of model outputs, prepared figures and
wrote main parts of the manuscript, created the website displaying all results, setup/calibrated the GR4J-lumped, HMETS-lumped, Blended-
lumped, and Blended-Raven models. **Hongren Shen** designed experiments, selected streamflow gauging stations, generated the routing prod-
uct for the Great Lakes and derived all basin characteristics, selected streamflow gauging stations, prepared input data, and setup/calibrated
the VIC-Raven model. **Bryan A. Tolson** co-managed the project, designed experiments, selected streamflow gauging stations and contributed
to outlining the manuscript. **Étienne Gaborit** prepared the geophysical fields and forcing files used by the regionally-calibrated models and
gave support in preparing the output files containing their auxiliary variables, helped review the Raven delineation setup, and setup/calibrated
the GEM-Hydro-Watroute and MESH-SVS-Raven models. **Richard Arsenault** provided the ERA5-Land data for snow water equivalent.
**James R. Craig** supported the setup/calibration of the GR4J-lumped, HMETS-lumped, Blended-lumped, and Blended-Raven models. **Vin-
cent Fortin** supported the setup/calibration of the GEM-Hydro-Watroute and MESH-SVS-Raven models. **Lauren M. Fry** supported the
setup/calibration of the LBRM-CC-lumped model. **Martin Gauch** setup/calibrated the LSTM-lumped model and gave feedback on the web-
site design. **Daniel Klotz** supported the setup/calibration of the LSTM-lumped model and gave feedback on the website design. **Frederik
Kratzert** supported the setup/calibration of the LSTM-lumped model and gave feedback on the website design. **Nicole O'Brien** supported
the setup/calibration of the WATFLOOD-Raven model. **Daniel G. Princz** supported the setup/calibration of MESH-SVS-Raven, MESH-





CLASS-Raven, and WATFLOOD-Raven. **Sinan Rasiya Koya** setup/calibrated the HYMOD2-lumped model. **Tirthankar Roy** supported the setup/calibration of the HYMOD2-lumped model. **Frank Seglenieks** supported the setup/calibration of the SWAT-Raven, WATFLOOD-Raven, and MESH-CLASS-Raven models. **Narayan K. Shrestha** setup/calibrated the SWAT-Raven and WATFLOOD-Raven models. **André G. T. Temgoua** setup/calibrated the MESH-CLASS-Raven model. **Vincent Vionnet** compared the ERA5-Land SWE reference dataset with SWE observations from the CanSWE dataset and provided the according codes. **Jonathan W. Waddell** setup/calibrated the LBRM-CC-lumped model.

*Competing interests.*  The authors declare that they have no conflict of interest.

*Acknowledgements.*  This research was undertaken thanks in part to funding from the Canada First Research Excellence Fund provided to the Global Water Futures (GWF) Project and the Integrated Modeling Program for Canada (IMPC). This study contains modified Copernicus Climate Change Service Information for ERA5-Land for the period 2000 to 2017 (accessed Oct 2021). The work was made possible by the facilities of the Shared Hierarchical Academic Research Computing Network (SHARCNET; www.sharcnet.ca) and Compute/Calcul Canada. Frederik Kratzert was partially supported by a Google Faculty Research Award. The Federal State of Upper Austria supports the ELLIS Unit Linz, the LIT AI Lab, and the Institute for Machine Learning where Martin Gauch, Daniel Klotz, and Frederik Kratzert are/were employed during this project. Martin Gauch was supported by the LIT DeepFlood project. All codes, examples, and data used for this study can be found on GitHub (https://github.com/julemai/GRIP-GL) upon publication. Note that the GitHub link will result in a 404-error if accessed without proper permission. The results of this study can be accessed through an interactive website (http://www.hydrohub.org/mips_introduction.html#grip-gl). This link will change once the manuscript is published and the webpage is moved to its final destination. This is NOAA Great Lakes Environmental publication number XXXX (will be assigned after acceptance of manuscript).



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
