# Peer review of "The Great Lakes Runoff Intercomparison Project Phase 4: The Great Lakes (GRIP-GL)"

_Hydrology and Earth System Sciences, 2022_

## Author Comment (AC1)

Dear Anonymous Referee #1,

thanks a lot for your positive feedback. We will reply below in detail to your comments. Your comments are formatted *italic*; our replies are highlighted **bold** and ***bold italic***. The line numbers in **red** are referring to the manuscript version you reviewed.

Thanks for your time and efforts evaluating our manuscript.

Best regards,
Juliane Mai and co-authors

*Mai et al. present a thorough model-intercomparison for the Great Lakes region.*

*The manuscript is very extensive, as is the way that the model intercomparison was managed. The intercomparison is conducted in a very structured manner and clearly was not opportunity-driven; teams had to create a new model-set up to be consistent with underlying data, and perform a new calibration. Also the analysis is very thorough and very honest, fairly comparing the performance of all the models based on different aspects. This is very much appreciated. It also demonstrates how much information can be gained from such a carefully designed experiment: many conclusions on different aspects can be drawn. This does make the manuscript quite long and has the risk that some conclusions might get lost in accompany of all the other conclusions, but the abstract provides a good summary.*

**We are really happy that the reviewer was so appreciative of our work and that the huge effort behind this project was conveyed in the manuscript.**

*I only have a few minor points:*

*From the methods-section it is not clear whether the LSTM was also trained with geographic data. Later I read it was, perhaps this can already be clarified earlier.*

**That's a good point. We will add more explicit information to the manuscript right when the LSTMs are introduced. The added text is highlighted *italic*; the rest is just slightly rearranged.**

**line 219-226: The LSTM setup used in this study is similar to that from Kratzert et al. [2019], and has been successfully applied for streamflow prediction in a variety of studies [e.g., Klotz et al., 2021, Gauch et al., 2021, Lees et al., 2021]. *The model inputs are nine basin-averaged daily meteorologic forcing data (precipitation, minimum and maximum temperature, u/v components of wind, etc.) as well as nine static scalar attributes derived for each basin from these forcings for the calibration period (mean daily precipitation, fraction of dry days, etc.), ten attributes derived from the common landcover dataset (fraction of wetlands, cropland, etc.), six attributes derived from the common soil database (sand, silt, and clay content, etc.), and five attributes derived from the common digital elevation map (mean elevation, mean slope, etc.).* The aforementioned data and attributes are based solely on the common dataset (Sect. 2.2). Streamflow is not part of the input variables. A full list of attributes used can be found in the Supplementary Material (Sect. S.2.1). The LSTM setup follows a global calibration strategy, which means that the model was trained for all 141 calibration stations at the same time, resulting in a single trained model for the entire study domain that can be run for any (calibration or validation) basin as soon as the required input variables are available. The LSTM training involved fitting around 300,000 model parameters. This number should, however, not be directly compared to the number of parameters in traditional hydrological models because the parameters of a neural network do not explicitly correspond to individual physical properties.**

*The donor-basin rule is indeed very basic... and as such I am wondering about the value of the space-validation. What does it mean when a model is good at simulating a catchment it hasn't "seen", with parameters based on another catchment? Does that make a model "better"? It could also just be an indication of how sensitive the output of this model is to different forcing / its own parameters, rather than a value-judgement of its performance. But this is just my thought.*

**We set out to empirically test how the developed models validate in space. Empirical evidence shows some models with better spatial validation performance and so, this means the aggregate model and model build decisions (model structure, model calibration and selected regionalization method taken together) cause these perfor-**

mance differences. So we agree that 'better' spatial validation performance does not strictly mean a better model in terms of model structure (e.g., SWAT better than WATFLOOD).

As such, we change the manuscript to emphasize that when interpreting spatial validation results, the 'model' is in fact the model structure/equations, the calibration approach and the regionalization strategy all combined. We suggest manuscript adjustments as follows (added text highlighted with italic font):

**line 463 ff:** The spatial validation (C; time period trained but location untrained) can be regarded to be more difficult especially for locally calibrated models given that one either needs a global/ regional model setup or a good parameter transfer strategy for ungauged/ uncalibrated locations. The spatio-temporal validation (D) can be regarded to be the most difficult validation experiment as both location and time-period were not included in model training. *The two latter validation experiments including the spatial transfer of knowledge provide an assessment of the combined quality of a 'model' which in this context includes model structure/equations, calibration approach, and regionalization strategy.*

Note that we have clearly stated the need to follow-up on improved regionalization approaches:

**line 755 ff:** Unless sophisticated parameter transfer methods are tested and employed, locally calibrated models are not well suited for simulations in ungauged areas, due to their lack of spatial robustness. The impact of more sophisticated donor-basin mapping methods will be evaluated in a follow-up study.

**line 1200 ff:** Future work will focus (among others) on testing more advanced donor basin mappings for the locally calibrated models, [...]

*It is appreciated that mistakes in the procedures are openly shared, such as about the PET-controlling constant for LBRM-CC calibration. However, there are no consequences related to this point. For instance, it is used as an argument to explain lower performance, but a lack of applying a constraint should actually result in better model performance because during the calibration there was more freedom to fit this parameter (or in equal model performance because the calibration algorithm did end up at the correct spot after all). The implications of this error for comparability are not clear. (same for the other calibration bug with SVS LSS)*

Insightful comment here - thanks for that. In our manuscript we need to be careful to always speak about performance and expected changes in performance with respect to a specific variable (streamflow or AET in this case). The AET bug in LBRM-CC surely explains poor AET performance. However, the reviewer is correct that impacts on streamflow performance of fixing this AET bug are unclear as streamflow could degrade if PET is more constrained. Hence, we acknowledge the implications of these errors are not clear and thus make changes in the manuscript to note the following: 1) We are clear that the expected results upon fixing the bugs are based on the individual modelling team expectations. 2) The actual implications of these bugs can only be assessed upon model recalibration with the bugs fixed. 3) We expect to be able to report on these in the future on the HydroHub website. We did not include revised model versions in the current study as this would have violated the blind validation concept of the study (models were calibrated once and afterwards no modification was allowed to guarantee a fair comparison).

We suggest the following changes in the manuscript (modifications highlighted in italic font):

**line 257 ff:** In addition, the LBRM-CC-lumped modelling team found that an improved representation of the long term average temperatures applied in the PET formulation improves *the AET simulations* in some tested watersheds (PET is a function of the difference between daily temperature and long term average temperature for the day of year). *The impact of these bug fixes on the performance regarding streamflow or other variables like evapotranspiration across the entire study domain is not yet clear and will need to be confirmed through recalibration of the model.*

**line 973 ff:** The LBRM-CC-lumped model has the overall weakest performance mostly due to its lower performance across the Ottawa River basin which is consistent with the weak performance of this model regarding actual evapotranspiration discussed earlier. Several avenues have been identified after posting the model results for LBRM-CC-lumped that may, *after recalibration,* improve the LBRM-CC-lumped performance in future studies.

**line 676 ff:** The main reason why GEM-Hydro-Watroute has a significantly lower performance than MESH-SVS-Raven, despite the former mainly relying on parameters calibrated with the latter, is  *believed* to be due to a bug that was present in the MESH-SVS-Raven model and related to the reading of vegetation cover from the geophysical files provided to the model. Note that this bug was not present in previous studies [Gaborit et al., 2017, Mai et al., 2021] as it was due to the specific NetCDF format used with MESH-SVS-Raven input/output files during this work. This led to the SVS LSS not using the right information for vegetation cover during calibration, and therefore to calibrated parameters that were not optimal for SVS inside GEM-Hydro-Watroute, where the reading of vegetation cover was done properly. *It is expected that when the bug is fixed and MESH-SVS-Raven is recalibrated, both MESH-SVS-Raven and GEM-Hydro-Watroute will exhibit better scores regarding the auxiliary evapotranspiration variable, and that GEM-Hydro-Watroute streamflow performances will be closer to the performances of MESH-SVS-Raven. A revised version of both models will be posted on HydroHub when this is done.*

In any case, the results of improved model version will then be made available on the website (`hydrohub.org`) if they are made available by the modelling teams. This is indicated in the manuscript:

**line 634 ff:** Additional models might be added at a later point to this website. *New calibration and validation results produced with revisions to these GRIP-GL models will be posted on HydroHub if the respective modelling teams decide to recalibrate their models.*

**line 683 ff:** However, new results for both models could be added to the website in the near future.

**line 1203 ff:** [...] adding additional models to the website once they become available, [...]

*It is nice that the majority of the models applied the same calibration algorithm, but all used slightly different settings. Was this determined based on expert judgment?*

Yes, it is. The models are very different in nature. Some run extremely fast and allow for more model evaluations and even several independent calibration trials from which the best was picked in the end. Other models have runtimes of several hours and can only be calibrated with smaller budgets to be feasible. In majority of cases, these

teams relied on a common algorithm based on their individual experiences using it in past studies. We did not enforce the algorithm choice on anyone. We did not want to restrict the models' performance by enforcing same budgets. The task for the experts was to provide the best model setup they can deliver with a given set of inputs. The rest was up to their judgment. We thank the reviewer for the question and will add the following statement to the manuscript (addition highlighted in *italic* font):

**line 208 ff:** The first group is the Machine-Learning based model which happens to be also the only model with a global setup (Section 2.4.1), the second group is comprised of the seven models that are locally calibrated (Section 2.4.2) and the third group is the five models that followed a regional calibration strategy (Section 2.4.3). *The calibration strategy (local, regional, global) and calibration setup (algorithm, objective, budget) was subject to expert judgment of each modeling team. The main goal of this project was to deliver the best possible model setup under a given set of inputs; the standardization and enforcement of calibration procedures would have limited this significantly due to the wide range of model complexity and runtimes.* The models are briefly described below including a short definition of these three calibration strategies.

*Some models were calibrated regionally, other locally. It is unclear why which models where used in one way or the other. I guess because this fits the general philosophy of this model / its common use. Maybe this can be clarified in Ch 2.*

This is correct; some models allow for a regional calibration because they are actually setup of entire domains and then evaluated at specific locations (streamflow gauging stations) while other models are only setup for the domains corresponding to exactly one streamflow gauging location. The latter can hence only be locally calibrated. However, models that can be setup for entire regions can also be calibrated locally only (as done, for example, for VIC-Raven). Models with regional and global setups usually have much longer runtimes and are therefore computationally more expensive to calibrate.

We thank the reviewer for highlighting that this might not be clear to the readers. We will add more information to the manuscript. Since it was nicely related to the previous comment, we will add the information at the same place (see reply to previous comment):

**line 208 ff:** The first group is the Machine-Learning based model which happens to be also the only model with a global setup (Section 2.4.1), the second group is comprised of the seven models that are locally calibrated (Section 2.4.2) and the third group is the five models that followed a regional calibration strategy (Section 2.4.3). *The calibration strategy (local, regional, global) and calibration setup (algorithm, objective, budget) was subject to expert judgment of each modeling team. The main goal of this project was to deliver the best possible model setup under a given set of inputs; the standardization and enforcement of calibration procedures would have limited this significantly due to the wide range of model complexity and runtimes.* The models are briefly described below including a short definition of these three calibration strategies.

*l. 487-490 (p19) unclear what is meant here.*

The paragraph referred to here is part of Sect. 2.6 on "Model evaluation setup and datasets":

**line 487-490:** It is not known if models have been trained in previous studies against

any of such data and, for example, model structures or process formulations might have been informed by such a preliminary training.

What is meant here, is that during the initial model development (potentially many of years ago), the model structure and process formulation of, for example, snow processes or evapotranspiration was informed by previous model evaluations regarding the same datasets. So, it might be that some models benefit from the fact that we picked exactly the datasets we picked and not, for example, station datasets for snow surveys or evapotranspiration.

We intend to replace the paragraph with the following hoping that this is more clear:

line 487-490: It is not known if models have been trained in previous studies against any of the observations we use for validation or model evaluation. Any model previously using such data to inform model structure or process formulations might have an advantage relative to another model whose structural development did not involve testing in this region.

*l. 604 (p23) not very clearly explained.. I guess it also depends on the shape of your pareto front (if it exists at all). It would be nice to see it somehow in a 2D-version (e.g. for two variables only), or a 3D version.*

The paragraph referred to here is part of Sect. 2.9 on "Multi-objective multi-variable model analysis":

line 603-604: When multiple models A, B, C, ..., M are under consideration, model A is dominated when there is at least one other model that dominates model A otherwise model A is non-dominated (e.g., no model is objectively superior to model A).

We apologize for the unclear explanation. We agree that the concept of non-dominance/ dominance in multi-objective analyses is challenging. We agree with the reviewer that a more detailed description and a visualization based on the reviewer's suggestion would be a helpful addition to the Supplementary Material. We therefore suggest adding the following subsection and figure to the Supplements just after the current subsection S.1:

line 25 ff (supplements): S.2 Visual depiction of the multi-objective multi-variable model analysis

The concept of non-dominance/ dominance in multi-objective analyses is used for the multi-objective multi-variate model analysis (Sec. 2.9 main manuscript). This refers to the classic definition of that concept which is independent of the shape of the pareto front. Fig. S2 is provided as a visual explanation of what it means that a model dominates other models (panel A), a model is dominated by another model (panel B), a model that is not dominated by other models (panel C), the entire set of non-dominated models (panel D) forming the Pareto front (panel E). We visualized this in 2D picking two objective functions (here KGE regarding streamflow and AET) for demonstration purposes. In the study itself only 3D or 4D pareto fronts were evaluated and reported on in Fig. 8 of the main manuscript. The 3D and 4D examples would, however, be harder to visualize intuitively. All these concepts can (mathematically) be applied for n-dimensional problems though.

We will refer to this material in the main manuscript as follows:

[Figure]

Figure S2: The concept of non-dominance and dominance of multi-objective problems. The example of a two-dimensional calibration problem is chosen. Both objectives (x-axis and y-axis) are assumed to be minimized and hence the "Utopia" point is located at the origin for simplicity. For demonstration purposes the first objective is chosen to be $1 - \mathrm{KGE_Q}$ where $\mathrm{KGE_Q}$ is the model performance regarding streamflow and the second objective is set to be $1 - \mathrm{KGE_{AET}}$ where $\mathrm{KGE_{AET}}$ is the model performance regarding actual evapotranspiration. The circle markers in each panel indicates one of the twelve models evaluated. (A) A model (dark gray marker) is dominating other models (light gray markers) if it is superior for all objectives. (B) A model is dominated by all models that are better in all objectives. (C) A model is non-dominated if it is not dominated by any model, i.e. all other models are worse in at least one of the objectives. (D) There might be several non-dominated models (red markers) which (E) form the so-called Pareto front (red line). To obtain results in Fig. 8 of the main manuscript the analysis is performed for each of the 212 catchments of the study. The number of times a model is part of the pareto front (red dot in panel E) is used as the measure in Fig. 8.

**line 606 ff:** **Theoretically, any number between 1 to M models can form the Pareto front.** *A visual depiction of Pareto fronts in a 2D example can be found in the Supplementary Material Sec. S.2 and Fig. S2.*

*The link to the website is now mentioned quite late. It is a very nice feature, would be nice if it would be mentioned earlier in the text.*

**The link is actually hidden in the reference [Mai, 2022] and therefore cited much earlier, i.e. line 88. But we fully agree that this is likely not the best way. We will adjust the first citations of this reference and add the link directly. For example:**

**line 86 ff:** **Note that this work is accompanied by an extensive supplementary materials document, primarily providing more details for model setups, and an interactive website (`http://www.hydrohub.org/mips_introduction.html`) [Mai, 2022] for sharing and exploring comparative results.**

*In the conclusion it is clearly stated that gridded evaluation might be preferred over basin evaluation (both could demonstrate different results). This is not mentioned as such in the abstract, where only the difference between the two is mentioned.*

**Great point. We will adjust the abstract as follows (adjustment is highlighted in *italic* font):**

**line 21 ff:** **(4) Comparisons of additional model outputs (AET, SSM, SWE) against**

gridded reference datasets show that aggregating model outputs and the reference dataset to basin scale can lead to different conclusions than a comparison at the native grid scale. *The latter is deemed preferable; especially* for variables with large spatial variability such as SWE.

**References**

Étienne Gaborit, Vincent Fortin, Xiaoyong Xu, Frank Seglenieks, Bryan Tolson, Lauren M Fry, Tim Hunter, François Anctil, and Andrew D Gronewold. A hydrological prediction system based on the SVS land-surface scheme: efficient calibration of GEM-Hydro for streamflow simulation over the Lake Ontario basin. Hydrology and Earth System Sciences, 21(9):4825–4839, 2017.

M. Gauch, F. Kratzert, D. Klotz, G. Nearing, J. Lin, and S. Hochreiter. Rainfall–runoff prediction at multiple timescales with a single long short-term memory network. Hydrology and Earth System Sciences, 25(4):2045–2062, 2021. doi: 10.5194/hess-25-2045-2021. URL `https://hess.copernicus.org/articles/25/2045/2021/`.

D. Klotz, F. Kratzert, M. Gauch, A. Keefe Sampson, J. Brandstetter, G. Klambauer, S. Hochreiter, and G. Nearing. Uncertainty estimation with deep learning for rainfall–runoff modelling. Hydrology and Earth System Sciences Discussions, 2021:1–32, 2021. doi: 10.5194/hess-2021-154. URL `https://hess.copernicus.org/preprints/hess-2021-154/`.

Frederik Kratzert, Daniel Klotz, Guy Shalev, Günter Klambauer, Sepp Hochreiter, and Grey Nearing. Towards learning universal, regional, and local hydrological behaviors via machine learning applied to large-sample datasets. Hydrology and Earth System Sciences, 23(12):5089–5110, 2019. doi: 10.5194/hess-23-5089-2019.

T. Lees, M. Buechel, B. Anderson, L. Slater, S. Reece, G. Coxon, and S. J. Dadson. Benchmarking data-driven rainfall–runoff models in great britain: a comparison of long short-term memory (lstm)-based models with four lumped conceptual models. Hydrology and Earth System Sciences, 25(10):5517–5534, 2021. doi: 10.5194/hess-25-5517-2021. URL `https://hess.copernicus.org/articles/25/5517/2021/`.

Juliane Mai. GRIP-GL interactive website. `http://www.hydrohub.org/mips_introduction.html#grip-gl`, 2022. Accessed: 2022-01-19.

Juliane Mai, Bryan A Tolson, Hongren Shen, Étienne Gaborit, Vincent Fortin, Nicolas Gasset, Hervé Awoye, Tricia A Stadnyk, Lauren M Fry, Emily A Bradley, Frank Seglenieks, André G T Temgoua, Daniel G Princz, Shervan Gharari, Amin Haghnegahdar, Mohamed E Elshamy, Saman Razavi, Martin Gauch, Jimmy Lin, Xiaojing Ni, Yongping Yuan, Meghan McLeod, Nandita B Basu, Rohini Kumar, Oldrich Rakovec, Luis Samaniego, Sabine Attinger, Narayan K Shrestha, Prasad Daggupati, Tirthankar Roy, Sungwook Wi, Time Hunter, James R Craig, and Alain Pietroniro. Great Lakes Runoff Intercomparison Project Phase 3: Lake Erie (GRIP-E). Journal of Hydrologic Engineering, 26(9):05021020, June 2021.

---

## Author Comment (AC2)

Dear Matteo,

thanks a lot for your positive feedback. We very much appreciate the time and effort you made to evaluate our manuscript. We happily reply below in detail to your comments. Your comments are formatted *italic*; our replies are highlighted **bold** and ***bold italic***. The line numbers in **red** are referring to the manuscript version you reviewed.

Many thanks,
Juliane Mai and co-authors

*The paper contributes a comprehensive model intercomparison across 13 hydrologic models, including Machine Learning based, conceptual, and physically based models. The analysis is run over the Great Lakes region looking at the model's ability to simulate streamflow, actual evapotranspiration, surface soil moisture, and snow water equivalent. The comparison is performed looking at simulated output aggregated to basin-scale as well as at grid-level, considering temporal and spatial validation. The study is extremely well designed and it provides a solid contribution to the existing literature. The manuscript is also well written and definitely interesting for HESS readership. I only have a few comments that I would recommend addressing before accepting the paper for publication.*

**Thanks so much for this positive evaluation of our manuscript. We are happy that the work is making a contribution to the existing literature. Thanks again for your time and effort to go through this extensive and long manuscript.**

1. *The model intercomparison reads extremely solid in terms of using consistent data, forcing, etc. as well as in terms of the adopted calibration-validation scheme. Yet, the description of the different models' calibrations in Section 2 seems to introduce quite some variability whose potential implications are not discussed. Although most models have been calibrated using the same algorithm, it is not clear whether the different modeling teams had some guidelines/constraints about the calibration effort to somehow harmonize it across models. Since it does not seem there was any limit on the number of model evaluations run (or on the total time spent) during the calibration, I'm wondering whether some results could be explained by better/worse calibration results. This aspect could also be an interesting finding of the analysis, but to be fair it should be derived by coordinating the calibration efforts. For example, the LSTM model involves 300,000 parameters, and being a data-driven model is by definition more flexible than the other models considered. This LSTM model was calibrated in 2.75 hours; how about the other models? Is the effort of running 300 iterations for calibrating the 9 parameters of the LBRM-CC-lumped model comparable?*

**Thanks for the comment. The models are very different in nature. Some run extremely fast and allow for more model evaluations and even several independent calibration trials from which the best was picked in the end. Other models have runtimes of several hours and can only be calibrated with smaller budgets to be feasible. In majority of cases, these teams relied on a common algorithm based on their individual experiences using it in past studies. We did not enforce the algorithm choice on anyone. We did not want to restrict the models' performance by enforcing same budgets. The task for the experts was to provide the best model setup they can deliver with a given set of inputs. The rest was up to their judgment. We thank the reviewer for the question and will add the following statement to the manuscript (addition highlighted in *italic* font):**

line 208 ff: **The first group is the Machine-Learning based model which happens to be also the only model with a global setup (Section 2.4.1), the second group is comprised of the seven models that are locally calibrated (Section 2.4.2) and the third group is the five models that followed a regional calibration strategy (Section 2.4.3).** ***The calibration strategy (local, regional, global) and calibration setup (algorithm, objective, budget) was subject to expert judgment of each modeling team. The main goal of this project was to deliver the best possible model setup under a given set of inputs; the standardization and enforcement of calibration procedures would have limited this significantly due to the wide range of model complexity and runtimes.*** **The models are briefly described below including a short definition of these three calibration strategies.**

*2. One of the key assumptions of the analysis is considering only streamflow gauges in low-human impacted watersheds. While the authors clearly motivate this choice, I believe the paper would benefit from some further elaboration around this point given the somehow limited number of "pristine" river basins worldwide, see for example Belletti et al. [2020]. Which type of bias we could expect in using these models in a human-impacted basin? Are these biases consistent across models, or can some categories better capture human inference even if not explicitly described? I believe this type of reasoning could be a good addition to the model discussion, which could be perhaps potentially supported by looking at the model performance in some sampled stations currently excluded from the analysis.*

**We are sorry that we caused a misunderstanding here. The basins we have picked are indeed NOT all low-human impact; only the basins classified under "Objective 1" are low-human impact (see for example Tab. S15 in the Supplementary Material). The table caption actually states "[...] The objective for each basin is assigned to be 1 if the watershed is of low-human impact while it is assigned to 2 if the gauge station is most downstream to one of the five lakes or the Ottawa River.[...]". All basins that are objective 2 and not objective 1 are considered to be not low-human impact.**

**In summary:**

**[station tagged as "objective 1" only]: The watershed is low-human impact and not most downsteam to one of the five lakes or the Ottawa River. There are 66-29=37 such calibration and 33-14=19 such validation stations.**

**[station tagged as "objective 1" and "objective 2"]: The watershed is low-human impact and most downsteam to one of the five lakes or the Ottawa River. There are 29 such calibration and 14 such validation stations.**

**[station tagged as "objective 2" only]: The watershed is most downstream to one of the five lakes or the Ottawa River but is not low-human impact. There are 104-29=75 such calibration and 52-14=38 such validation stations.**

**The numbers of stations in each of the three categories listed above are already given in the caption of figure 1:**

**Caption Fig. 1:** **Panel A shows the location of stations used for calibration regarding stream- flow: 66 of them are downstream of a low-human impact watershed (objective 1; large black dots) and 104 stations are most downstream draining into one of the five lakes or the Ottawa River (objective 2; smaller dots with white center). In total, there are 141 stations used for calibration as 29 stations are both low-human impact and most downstream (large black dots with white center; 141 = 66 + 104 - 29). Panel B shows the 71 validation stations of which 33 are low human impact, 52 are most downstream and 14 are both low human impact as well as most downstream (71 = 33 + 52 - 14).**

**We suggest to make the description of the objectives and the distinction of the three cases more clear in the manuscript (suggested addition highlighted in italic font):**

**line 441 ff:** **[...] streamflow gauges need to be either downstream of a low-human impact watershed (objective 1) or most downstream of areas draining into one of the five Great Lakes or into the Ottawa River (objective 2). *If a watershed is most downstream and human impacts are low, the station would hence be classified as both objective 1 and 2.* Objective 1 was defined to give all models – especially the ones without the possibility to account for watershed management rules –**

to perform well. Objective 2 was chosen since the ultimate goal of  *many operational* models *in this region* is to estimate the flow into the lakes (or the Ottawa River). *This classification of gauges was a part of the study design that ends up not being evaluated in this paper. This is because each of the modelling teams decided to build their models using all Obj. 1 and Obj. 2 stations treated the same way. As such our results do not distinguish performance differences for these two station types. The information is included here so follow-up studies by our team and others can evaluate this aspect of the results.*

3. *The temporal validation of the models is based on model simulations over the period 2011-2017, with the models calibrated over 2001-2010. This looks certainly good, but I was then expecting the authors to somehow comment/discuss the role of nonstationary forcing as I expect that data (e.g. temperature) could show some trends over these 17 years. If this is the case, how did you handle such trends? Were the data de-trended or did you use the raw observations? Moreover, what are the authors' recommendations for developing hydrologic models under such evolving conditions? Again, are there any class of models more prone/robust to possible extrapolation biases induced by global warming?*

We indeed know that the calibration period (2000-2010) is a dry period while the validation period (2011-2017) is known to be very wet. This is reported, for example, here: `https://www.lre.usace.army.mil/Portals/69/docs/GreatLakesInfo/docs/UpdateArticles/update206.pdf?ver=2020-07-01-115844-313`. We suggest adding the following to the manuscript:

**line 440 ff:** It is known that the calibration period (2001-2010) is a dry period while the validation period (2011-2017) is known to be very wet [US Army Corps of Engineers: Detroit District, 2020]. This might have an impact on model performances - especially in temporal validation experiments. In this study no specific method has been applied to account for these trends in the meteorologic forcings.

This is certainly something that one might want to take into account while model building and training. We however did not. The very good performance of the LSTM-lumped model in temporal validation (median KGE of 0.82 compared to median KGE of 0.97 in calibration; Fig. 3C vs. Fig. 3A) shows that it might not even be necessary in order to achieve good performance. The larger impact on model performance has a transfer in space (rather than time) as can be seen in spatial validation (median KGE of 0.76 compared to median KGE of 0.97 in calibration; Fig. 3B vs. Fig. 3A).

Essentially, all these questions/suggestions are good but were not addressed and are out of scope for us. They might be worth a future inspection to see if the model performance could be improved by de-trending the forcings; a longer forcing dataset might be required though.

4. *Lastly, as the authors probably know the paper is quite lengthy and it does require substantial commitment to get to the end. I think the authors did already a good effort in guiding readers using a good structure and providing summaries of each section, but I would suggest - if feasible - to further shortening the paper in order to facilitate a complete read. I don't have clear recommendations on how to do this; perhaps an idea could be to move the model description of section 2.4 into an appendix keeping only a summary in the main text?*

We totally agree with the reviewer that this is a very long manuscript. We were hoping that the clear structure and brief description of subsections at the beginning

of each section would be helpful for the reader to navigate through the manuscript and directly skip to the sections of interest. We tried to keep the sections as brief as possible without removing any detail that is required to follow the main analyses and conclusions. We do not want to move the (brief) model descriptions to the Supplements as this seems integral for a model intercomparison study. All details that are not regarding the following specifics have been already moved to the Supplements (the following list of bullet points was given to each collaborator contributing a section describing their model):

- introductory sentence including major references for model and aim of model

- model resolution (spatial, temporal)

- used forcings and derived basin attributes

- calibration method (algorithm, objective, iterations/budget, independent trials, parameters go to Supplements)

- validation: donor basin mapping or regional/global setup?

- evaluation: model outputs AET, SSM, SWE (maybe add the actual model variable that is dumped to the output file; make note about SSM how this is modeled and why it was important to show only standardized SSM (i.e., only using correlation))

We hope it is ok if we do not make any adjustments to the manuscript here.

**References**

Barbara Belletti, Carlos Garcia de Leaniz, Joshua Jones, Simone Bizzi, Luca Börger, Gilles Segura, Andrea Castelletti, Wouter van de Bund, Kim Aarestrup, James Barry, Kamila Belka, Arjan Berkhuysen, Kim Birnie-Gauvin, Martina Bussettini, Mauro Carolli, Sofia Consuegra, Eduardo Dopico, Tim Feierfeil, Sara Fernández, Pao Fernandez Garrido, Eva Garcia-Vazquez, Sara Garrido, Guillermo Giannico, Peter Gough, Niels Jepsen, Peter E Jones, Paul Kemp, Jim Kerr, James King, Małgorzata Łapińska, Gloria Lázaro, Martyn C Lucas, Lucio Marcello, Patrick Martin, Phillip McGinnity, Jesse O'Hanley, Rosa Olivo del Amo, Piotr Parasiewicz, Martin Pusch, Gonzalo Rincon, Cesar Rodriguez, Joshua Royte, Claus Till Schneider, Jeroen S Tummers, Sergio Vallesi, Andrew Vowles, Eric Verspoor, Herman Wanningen, Karl M Wantzen, Laura Wildman, and Maciej Zalewski. More than one million barriers fragment Europe's rivers. Nature, 588 (7838):436–441, December 2020.

US Army Corps of Engineers: Detroit District. Great Lakes Update – Volume 206: From Record-Lows to Record-Highs in 6 years. https://www.lre.usace.army.mil/Portals/69/docs/GreatLakesInfo/docs/UpdateArticles/update206.pdf?ver=2020-07-01-115844-313, 2020. Accessed: 2022-06-01.

---

## Author Comment (AC3)

Dear Dr. Ding,

thanks a lot for your feedback. We will reply below to your comments. Your comments are *italic*; our replies are highlighted **bold**. The line numbers in red are referring to the manuscript version you commented on.

Best regards,
Juliane Mai and co-authors

*Comparing a LSTM model and a one-step-ahead river forecast model*

*The study team reaches a most profound conclusion that the Machine Learning LSTM-lumped model outperforms 12 other physically based models for Great Lakes - Ottawa River region (Lines 14-16, Abstract, Main Result (1)).*

*I'm curious how the LSTM model (Sect. 2.4.1, S.2.1) compare with a simple one-step-ahead forecast model, AR(2), a second-order autoregressive process of the streamflow (only). The latter is constructed as follows (e.g., [Ding, 2018]):*

$$Q[t+1] = 0 + O[t] + (O[t] - O[t-1]) = 2O[t] - O[t-1] \tag{1}$$

*in which:*

*$O[t]$ and $Q[t]$ are the observed and simulated discharge, respectively, at current timestep t. This pre-defined AR(2) has a constant of zero, and lag 1 and 2 coefficients of 2 and -1, thus having a fixed variance for an observed hydrograph.*

*A comparison between the two on one of their study watersheds will help demonstrate their performances. One candidate could be Gauge ID 02GA047 - Speed River at Cambridge which has been used for validation purposes (Table S15). This is located on the east side of the Grand River, Ontario, opposite to the University of Waterloo campus, and has a drainage area of 782 sq. km.*

*On the Speed River, does a LSTM model that has been calibrated globally for the Great Lakes-Ottawa River region outperform an AR(2) too?*

**This is an out of scope analysis. All models were not allowed to use streamflow on day t-1 as an input variable to predict streamflow on day t. Perhaps this reviewer mistakenly thought the LSTM uses the previous day streamflows to predict streamflow for the current day. This is explicitly stated as not being the case in Section 2.4.1:**

**line 225:** **Streamflow is not part of the input variables.**

**The study design called for all models to utilize the same set of geospatial and forcing inputs to build the models, hence adding a new model with a new input variable not used by other models is completely inconsistent with our inter-comparison study design. A one day autoregressive model is incapable of simulation beyond the one day time horizon; these models are being assessed in their skill in simulating streamflow throughout a 7-year validation period. Future work by others can assess such a time series model against our suite of GRIP-GL models.**

**References**

John Ding. Interactive comment on "On the choice of calibration metrics for "high flow" estimation using hydrologic models" by Naoki Mizukami et al. Hydrol. Earth Syst. Sci. Discuss., 2018. URL https://doi.org/10.5194/hess-2018-391-SC1.